



Source attribution of Arctic black carbon constrained by aircraft and surface measurements

Junwei Xu[1], Randall V. Martin[1,2], Andrew Morrow[1], Sangeeta Sharma[3], Lin Huang [3], W. Richard Leaitch[3], Julia Burkart[4], Hannes Schulz[5], Marco Zanatta[5], Megan D. Willis[4],

Daven K. Henze[6], Colin J. Lee[1], Andreas B. Herber[5], Jonathan P.D. Abbatt[4]

[1]Department of Physics and Atmospheric Science, Dalhousie University, Halifax, NS, Canada

[2]Harvard-Smithsonian Center for Astrophysics, Cambridge, MA, USA

[3]Atmospheric Science and Technology Directorate/Science and Technology Branch,

Environment and Climate Change Canada, Toronto, Ontario, Canada

[4]Department of Chemistry, University of Toronto, Toronto, Canada

[5]Alfred Wegener Institute, Helmholtz Centre for Polar and Marine Research, Bremerhaven, Germany

[6]Department of Mechanical Engineering, University of Colorado, Boulder, CO, USA

Abstract

Black carbon (BC) contributes to both degraded air quality and Arctic warming, however sources of Arctic BC and their geographic contributions remain uncertain. We interpret a series of recent airborne and ground-based measurements with the GEOS-Chem global chemical transport model and its adjoint to attribute the sources of Arctic BC. The

springtime airborne measurements performed by the NETCARE campaign in 2015 and the PAMARCMiP campaigns in 2009 and 2011 offer BC vertical profiles extending to >6 km across the Arctic and include profiles above Arctic ground monitoring stations. Long-term ground-based measurements are examined from multiple methods (thermal, laser incandescence and light absorption) at Alert (2011-2013), Barrow (2009-2015) and Ny-

Ålesund (2009-2014) stations. Our simulations with the addition of gas flaring emissions are consistent with ground-based measurements of BC concentrations at Alert and Barrow in winter and spring (rRMSE < 13%), and with airborne measurements of BC vertical profile across the Arctic (rRMSE=17%).




Sensitivity simulations suggest that anthropogenic emissions in eastern and southern Asia are the largest source of the Arctic BC column both in spring (56%) and annually (37%), with larger contributions aloft than near the surface (e.g. a contribution of 66% between 400-700 hPa and of 46% below 900 hPa in spring). Anthropogenic emissions

from northern Asia contribute considerable BC to the lower troposphere (a contribution of 27% in spring and of 43% annually below 900hPa). Biomass burning has a substantial contribution to Arctic BC below 400 hPa of 25% annually, despite minor influence in spring (<10%).

Surface BC is largely influenced by anthropogenic emissions in winter and spring, and by

biomass burning in summer. At Alert and Barrow, anthropogenic emissions from northern Asia are the largest source of BC (>50%) in winter and those from eastern and southern Asia are the largest in spring (~40%). At Ny-Ålesund, anthropogenic emissions from Europe (~30%) and northern Asia (~30%) are major sources in winter and early spring. Biomass burning from North America is the most important contributor to

surface BC at all stations in summer, especially at Barrow where North American biomass burning contributes more than 90% of BC in July and August.

Our adjoint simulations indicate pronounced spatial and seasonal heterogeneity in the contribution of emissions to the Arctic BC column concentrations with noteworthy contributions from emissions in eastern China (15%) and western Siberia (6.5%).

Although uncertain, gas flaring emissions from oilfields in western Siberia could have a striking impact (13%) on Arctic BC loadings in January, comparable to the total influence of continental Europe and North America (6.5% each in January).

1. Introduction

The Arctic has warmed rapidly over the last few decades at a rate about twice the global

mean (AMAP, 2011; AMAP, 2015). By directly absorbing solar radiation, black carbon (BC) contributes substantially to the warming, impacting the Arctic in multiple ways (Bond et al., 2013; Flanner et al., 2007; Ramanathan and Carmichael, 2008; Sand et al., 2016; Shindell and Faluvegi, 2009). Near-surface BC particles over a highly reflective



surface (i.e. snow and ice in the Arctic) warm the atmosphere, and subsequently the surface (Quinn et al., 2008; Shaw and Stamnes, 1980). BC particles well above the surface warm the layer in which they reside and increase the stability of the Arctic atmosphere (e.g. Brock et al., 2011). Deposition of BC onto snow and ice can reduce

surface albedo and enhance light absorption by snow and ice (Chýlek et al., 1983; Wiscombe and Warren, 1980), and trigger chain reactions involving the acceleration of snow aging (Clarke and Noone, 1985; Hansen and Nazarenko, 2004), both leading to accelerated melting. (Namazi et al., 2015; Quinn et al., 2008). The modified local radiative balance exerted by deposited BC has the potential to further affect climate at a

larger scale (Doherty et al., 2010; Flanner et al., 2007). However, the contribution of BC to changes in the Arctic is highly uncertain, partly due to large uncertainties in sources of Arctic BC. Additional interpretation of BC observations in the Arctic is therefore needed to understand its sources.

Surface observations of BC have been intensively conducted at several Arctic locations

over the past few decades (Delene and Ogren, 2002; Eleftheriadis et al., 2009; Sharma et al., 2006; Yttri et al., 2014), and many studies have found that the observed surface BC in the Arctic is primarily transported from source regions outside the Arctic (e.g. Huang et al., 2010; Liu et al., 2015; Wang et al., 2011; Winiger et al., 2016; Yttri et al., 2014). Early observations have identified anthropogenic emissions in northern Eurasia as the

primary source of BC in the Arctic by analyzing the characteristics of chemical tracers (Lowenthal et al., 1997; Lowenthal and Kenneth, 1985). East and South Asia were excluded from these early studies because they were assumed to be unlikely sources due to the long distance (Cheng et al., 1993; Rahn, 1981). However, Koch and Hansen (2005) suggested that East and South Asia were comparable to Russia and Europe as

sources to the Arctic surface BC, and were dominant sources of BC in the upper troposphere. Subsequent studies supported the importance of East and South Asia to the Arctic upper troposphere (Sharma et al., 2006; Shindell et al., 2008; Wang et al., 2011). Shindell et al. (2008) studied the sensitivity of Arctic BC concentrations to perturbations in emissions in each region using results from a coordinated model



intercomparsion, and found that East and South Asia were indeed dominant sources in the Arctic upper troposphere, but at the surface, Europe remained the predominant contributor. Sharma et al. (2013) also found that East Asia had little influence at the surface but contributed substantially to atmospheric Arctic BC burden in winter. Recent

work by Stohl et al. (2013) raises questions about these prior studies by identifying the importance seasonally varying residential heating and suggesting that gas flaring in high-latitude regions is a significant overlooked source of Arctic surface BC that remains missing from most inventories (e.g. Bond et al., 2007 inventory). Sand et al. (2016) found that Russian flaring emissions make the second largest contribution to the

warming of Arctic surface temperature following Asian domestic emissions. Furthermore, evidence is emerging that the BC observations to which many prior modeling studies compared may have been biased by 30% (Sinha et al., accepted) or a factor of 2 (Sharma et al., submitted). Additional attention is needed to these issues.

In addition to anthropogenic emissions, another important periodic source of BC is

biomass burning (Lavoué et al., 2000). For example, several simulations suggest that a severe air pollution episode in the European Arctic in 2006 spring (Stohl et al., 2007), and a strong increase in BC concentrations at four Arctic monitoring stations in summer 2004 are all attributable to intense biomass burning events in northern Eurasia and North America (Stohl et al., 2006). Subsequent studies support the large contributions of

biomass burning to Arctic BC concentrations (Evangeliou et al., 2016; Warneke et al., 2009; Yttri et al., 2014), yet disagree quantitatively. Warneke et al. (2009) suggested that biomass burning contributed at least 80% to the Arctic atmospheric BC burden in April 2008, whereas Wang et al. (2011) indicated that biomass burning contributed 50% of total BC in the Arctic tropospheric column during the same period. Evangeliou et al.

(2016) found the contribution of biomass burning to Arctic surface BC as site-dependent, annually contributing 71% to surface BC at Alert, compared to 47% at Barrow. Additional interpretation of observations is needed to constrain this uncertain source.





As implied previously, emissions in mid- and low-latitude regions affect not only the surface of the Arctic, but also column loadings and vertical distributions (Koch and Hansen, 2005; Sharma et al., 2013; Shindell et al., 2008; Wang et al., 2011). Since the latter has important implications for radiative forcing (Koch et al., 2009; Samset et al.,

2014; Samset and Myhre, 2011), it is also crucial to identify significant sources and quantify their contributions to the vertical distribution and column concentrations of BC in the Arctic. However, vertical profiles in the Arctic have been scarce (Brock et al., 2011; Jacob et al., 2010). The NETCARE (Network on Climate and Aerosols: Addressing Key Uncertainties in Remote Canadian Environments, http://www.netcare-project.ca)

aircraft campaign in 2015 and the PAMARCMiP (Polar Airborne Measurements and Arctic Regional Climate Model Simulation Project) aircraft campaigns in 2009 and 2011 offer a new dataset of BC measurements across the Arctic. All three campaigns were performed in spring when BC is most abundant, and traveled along similar routes across the entire western Arctic and near long-term ground monitoring stations in the Arctic

(Alert, Barrow and Ny-Ålesund). Airborne measurements during all three campaigns were performed with the state-of-the-science single particle soot photometer (DMT-SP2; Stephens et al., 2003; Stone et al., 2010) for the measurement of refractory BC (rBC) mass concentrations. The spatial and multi-year coverage of airborne measurements during these campaigns offer comprehensive insight into BC

distributions and high representativeness of characteristics of Arctic BC.

Source attributions of pollution in the Arctic are commonly estimated by back-trajectory analysis that identifies possible source regions by tracking air mass flow (Barrett et al., 2015; Harrigan et al., 2011; Huang et al., 2010; Liu et al., 2015), and by chemical transport models that a perturbation is applied to emission sources and then compares

to an unperturbed run to infer the influence of emissions on the simulation (Evangeliou et al., 2016; Fisher et al., 2010; Koch and Hansen, 2005; Mungall et al., 2015; Sharma et al., 2013). These traditional approaches have been insightful, but suffer from coarse regional estimates of the source location. The adjoint of a global chemical transport model (Henze et al., 2007) efficiently determines the spatially resolved source



contribution to receptor locations by calculating the gradient of a cost function (e.g. Arctic column BC concentrations) with respect to the perturbations of the initial conditions (e.g. emissions). This approach has been applied in previous studies to identify origins of BC arriving at the Himalayas and Tibetan Plateau (Kopacz et al., 2011)

and to quantify source contributions to Arctic surface BC in April 2008 (Qi et al., 2017a). We extend the application of this method to investigate responses of Arctic BC column concentrations to changes in local emissions.

In this study, we first evaluate the BC concentrations simulated with the GEOS-Chem model with surface and aircraft measurements in the Arctic in order to assess the

quality of different emission representations. Then sensitivity simulations are conducted to assess the regional contributions to the observed BC in the Arctic. We subsequently use the adjoint of the GEOS-Chem model to investigate the spatially resolved sensitivity of Arctic BC column concentrations to global emissions.

2. Method

2.1 Surface measurements of BC in the Arctic

Surface BC mass concentrations were measured at three Arctic stations: Alert (Nunavut, Canada; 62.3° W, 82.5° N), Barrow (Alaska, USA; 156.6° W, 71.3° N) and Ny-Ålesund (Svalbard, Norway; 11.9° E, 78.9° N). Station locations are shown in Fig. 1. Following the recommendations of Petzold et al. (2013), measurements of BC based on light

absorption are here referred to as equivalent BC (EBC); measurements based on a laser induced incandescence technique (e.g. SP2) are referred to as refractory BC (rBC); and measurements based on a thermal volatilization in an oxygen-enriched environment are referred to as elemental carbon (EC).

EBC mass concentrations derived from an AE-31 Aethalometer (Magee Scientific Inc.) at

Alert are obtained from Environment and Climate Change Canada and those at Barrow and Ny-Ålesund are obtained from the EMEP and WDCA database (http://ebas.nilu.no/). The Aethalometer measures the absorption of light at 880 nm transmitted through





particles that accumulate on a quartz fiber filter and relate the change of light absorption to light absorption coefficients ($\sigma_{ap}$) using Beer's Law. EBC mass concentrations are derived from $\sigma_{ap}$ by adopting a mass absorption cross-section (MAC) of 16.6 $m^2 g^{-1}$ at all stations. This MAC value is recommended by the manufacturer for

Model AE31 at 880 nm. No scattering corrections have been applied to the EBC measurements from the Aethalometer.

EBC mass concentrations are also derived from a particle soot absorption photometer (PSAP, Radiance Inc.) that operates on a similar principle to the Aethalometer at the three stations. PSAP measures the absorption of light at 530 nm. $\sigma_{ap}$ data at Alert is

obtained from Environment and Climate Change Canada, and $\sigma_{ap}$ data at Barrow and Ny-Ålesund are obtained from the EMEP and WDCA database (http://ebas.nilu.no/). $\sigma_{ap}$ has been corrected for scattering following Bond et al. (1999) and is further reduced by 30% at all stations following Sinha et al. (accepted). $\sigma_{ap}$ values less than the detection limit (0.2 $Mm^{-1}$) are excluded. Recent evidence is emerging that the MAC is lower than

the traditional value of 10 $m^2 g^{-1}$, with recent effective MAC values ranging from 8 $m^2 g^{-1}$ (Sharma et al., submitted) to 8.7 $m^2 g^{-1}$ (Sinha et al., accepted). We adopt the average of these two values (8.4 $m^2 g^{-1}$) for application to PSAP measurements at all three sites.

Two additional measurements of BC mass concentrations are available at Alert: rBC and EC. rBC is measured via laser induced incandescence technique by an SP2 instrument

(Droplet Measurement Technologies Inc., Boulder, CO). The SP2 uses a high intensity laser (Ni-YAG) operating at 1064 nm of wavelength to selectively heat the individual particle up to 4000K. At such high temperature, the non-refractory components are evaporated and rBC mass is proportional to the intensity of the emitted incandescence light. The incandescence signal is calibrated using Aquadag particles of known size

selected with a differential mobility analyzer (Sharma et al., submitted). The detection range of the SP2 at Alert spans approximately between 75 nm and 530 nm volume-equivalent diameter (Sharma et al., submitted), assuming a rBC density of 1.8 $g cm^{-3}$ (Bond and Bergstrom, 2006). A lognormal function fit over the range of 80-225 nm is





applied to calculate rBC concentrations over the 40-1000 nm size range that increases the rBC concentrations by about 50 % (Sharma et al., submitted).

EC measurements at Alert are inferred from weekly-integrated samples of particles collected on quartz filters with a 1µm upper size cut and analyzed using an in-house
thermal technique referred to as EnCan-total-900 (Huang et al., 2006). The EnCan-total-900 method has three temperature steps with different redox conditions: 550°C and 870°C under pure helium and 900°C under helium + 10% oxygen. The retention times are 600 seconds at 550°C for OC, 600 seconds at 870°C for pyrolysis of OC and carbonate carbon, and 420 seconds at 900°C for EC. The 870°C pure helium step
releases pyrolysis OC and carbonate carbon to minimize the effect of OC charring on EC.

## 2.2 Aircraft measurements of BC in the Arctic

Aircraft measurements are obtained from a series of recent campaigns that offer new measurements in the lower troposphere across the Arctic. The PAMARCMiP campaigns conducted spring time surveys of sea ice thickness, aerosol and meteorological
parameters along the coast of the western Arctic onboard the Alfred Wegener Institute (AWI) Polar 5 aircraft. Data from two campaigns in April 2009 (Stone et al., 2010) and March 25[th] – May 6[th] 2011 (Herber et al., 2012) are used here. The NETCARE campaign in April 2015 continued and extended the PAMARCMiP campaigns observations using the AWI aircraft 6. Flight tracks of each campaign are shown in Fig. 1.

rBC concentrations were all measured with SP2 instruments (Droplet Measurement Technologies Inc., Boulder, CO) during the three campaigns. The SP2 used during the PAMARCMiP campaigns was previously described in Stone et al. (2010). The NETCARE 2015 campaign used the AWI's 8-channel SP2 with a detection range of 75 – 700 nm of volume-equivalent diameter (assuming a particle density of 1.8 g cm$^{-3}$) without
corrections for particles outside the size range. The incandescence signal was calibrated with particles of Fullerene soot size selected with a differential mobility analyzer.

## 2.3 Simulations of Arctic BC



We use the GEOS-Chem global chemical transport model (version 10-01; http://geos-chem.org/) and its adjoint (version 35) to simulate Arctic BC concentrations and their sensitivities to local emissions.

Figure 1 shows the annual mean BC emissions in our GEOS-Chem simulation averaged
over 2009, 2011 and 2015. We develop the simulation here to use global anthropogenic emissions of BC from version 2 of the HTAP (Hemispheric Transport of Air Pollution; http://www.htap.org/) emission inventory for 2010 (Gilardoni et al., 2011; Janssens-Maenhout et al., 2015) with regional overwrites over the United States (NEI 2011) for the most recent year (2011). The HTAP inventory is a compilation of different official
emission inventories from MICS-Asia, EPA-US/Canada and TNO-Europe data, gap-filled with global emission data of EDGARv4.1. The HTAP contains BC emissions by all major sectors, including energy and industrial production, transport and residential combustion.

Figure 2 shows annual HTAP BC emissions and its seasonal variation over the Arctic and
the Northern Hemisphere. The Bond et al. (2007) emission inventory for 2000 is included for comparison, since it has been widely used in modeling studies of Arctic BC (Koch et al., 2009; Liu et al., 2011; Shindell et al., 2008; Wang et al., 2011; Qi et al., 2017a; Qi et al., 2017b). The Bond et al. (2007) inventory is based on energy consumption in 1996 and contains similar emission sectors as in the HTAP. The HTAP
annual emissions over the Northern Hemisphere exceed those in Bond et al. (2007) by 30%, with a substantial difference in China and India where HTAP emissions are doubled compared to the emissions from Bond et al. (2007). A considerable increase of global energy consumption since 2001 especially in China and India may contribute to the difference (Li et al., 2015; Zhang et al., 2009). Both inventories have low BC emissions
within the Arctic. Fig. 2 also shows the seasonal variation of HTAP emissions that are high in winter and spring and low in summer over the Northern Hemisphere, owing to the seasonal variation of emissions from residential heating in the HTAP. Bond et al. (2007) emissions are non-seasonal.





We also include additional BC emissions from gas flaring in the oil and gas industry taken from version 5 of the ECLIPSE (Evaluating the climate and Air Quality Impacts of short-Lived Pollutants) emission inventory (Klimont et al., 2016; http://eclipse.nilu.no). Gas flaring emissions of BC are calculated based on gas flaring volumes developed within the

Global Gas Flaring Reduction initiative (Elvidge et al., 2007, 2011) and emission factors derived on the basis of particulate matter and soot estimates from CAPP (2007); Johnson et al. (2011) and US EPA (1995). Despite the small percentage (~5%) of flaring in total anthropogenic BC emissions over the Northern Hemisphere, flaring from Russia alone accounts for 93% of total anthropogenic BC emissions within the Arctic in the

ECLIPSE inventory.

Emissions from biomass burning are calculated from the GFED4 (Global Fire emissions Database version 4) inventory (Giglio et al., 2013). The GFED4 combines satellite information on fire activity and vegetation productivity to estimate globally gridded monthly burned area (including small fires) and fire emissions. We use emissions for

2009, 2011 and 2014 (the most recent year available) for the simulations of 2009, 2011 and 2015.

As discussed in section 2.1, measurements of BC depend on the analysis method. However, it is ambiguous what analysis method is used to derive BC emission factors or BC speciation factors in particulate matter in various emission inventories (Bond et al.,

2013). Therefore, we directly compare simulated BC concentrations with the best estimate of measured atmospheric BC.

The simulation of BC in GEOS-Chem is described in Park et al. (2003). BC emitted from all primary sources is in hydrophobic and hydrophilic states with a constant conversion time of one day. BC aerosols are removed through dry deposition as described in Zhang

et al. (2001) and Fisher et al. (2010). Hydrophilic BC aerosols are also removed through wet deposition following Liu et al. (2001).



Our GEOS-Chem simulations are driven by Modern-Era Retrospective Analysis for Research and Applications (MERRA) meteorological fields from the NASA Global Modeling and Assimilation Office (GMAO) at 2° × 2.5° spatial resolution with 47 vertical levels from the surface to 0.01hPa. We conduct simulations for 2009, 2011 and 2015

with a 10-minute operator duration for transport and a 20-minute operator duration for chemistry as recommended by Philip et al. (2016). We initialize the model with a 6-month spin-up before each simulation to remove the effects of initial conditions on aerosol simulations.

We conduct sensitivity simulations using the GEOS-Chem model to quantify the

contributions of regional emissions to Arctic (refer to the region north of 66.5 °N hereafter) BC concentrations by excluding the regional anthropogenic source. Regions are North America (180° W-50° W, 0° N – 80° N), Europe (50° W- 50° E, 30° N – 80° N), eastern and southern Asia (50° E – 150° E, 0° N – 50° N) and northern Asia (50° E – 180° E, 50° N – 80° N), as outlined in Fig.1. We also conduct sensitivity simulations to quantify

the contribution of biomass burning from North America and from the rest of the world to Arctic BC concentrations. These simulations are initialized with a 6-month spin-up as well.

We also apply the GEOS-Chem adjoint model to quantify the spatially resolved sensitivity of Arctic BC column concentrations to local emissions. A detailed description

of the adjoint model is given in Henze et al. (2007). Here we briefly describe the concept in the context of our study. The adjoint model offers a computationally efficient approach to calculate the sensitivity of a model output scalar, the cost function, to a set of model input parameters such as emissions. In this study, we define the cost function as the column concentrations of BC north of 66.5° N. The adjoint model calculates the

partial derivatives of this cost function with respect to the modeled atmospheric state in each model grid box at each time step. This calculation is performed iteratively backward in time through transport toward emissions to yield the sensitivity of the cost function with respect to emissions.





Our adjoint simulation is driven by GEOS-5 meteorology at 2° × 2.5° spatial resolution with 47 vertical levels from the surface to 0.01hPa for 2011. Differences between MERRA meteorological fields that are used in the forward model and GEOS-5 meteorological fields that are used in the adjoint are negligible ($r^2$ = 0.99 for Arctic

column BC concentrations for 2011) in the simulation of BC. Although the adjoint simulation is based on an earlier version (v8) of the GEOS-Chem model than the forward model version (v10-01) used in this study, it is found that the difference in BC concentrations at Arctic stations that are simulated with the adjoint and with the forward model is within 15% (Qi et al., 2017a).

2.4 Statistics

To assist with the evaluation of simulations, we define root mean square error (RMSE) and relative root mean square error (rRMSE) as

$$\text{RMSE}=\sqrt{\frac{1}{N}\sum_{i=1}^{N}(C_m(i) - C_o(i))^2} \qquad (1)$$

$$\text{rRMSE}=100\% \times \frac{\text{RMSE}}{\frac{1}{N}\sum_{i=1}^{N} C_m(i)} \qquad (2)$$

where $C_m(i)$ is the model simulated concentration and $C_o(i)$ is the measurement concentration. N is the number of measurements.

3. Results

3.1 Evaluation of GEOS-Chem simulated BC concentrations in the Arctic

Figure 3 shows the seasonal variation of BC concentrations from measurements and

simulations at Alert, Barrow and Ny-Ålesund stations. Different black lines indicate different instruments. Slight differences exist in sampling periods from different instruments. Restricting measurements to common years (e.g. 2010-2014 of Barrow) changes monthly means by less than 13%, except for a 40% change at Ny-Ålesund in April that arises from limited data coverage in common years (measurements halved for





common years). At Alert, a diversity of instruments offers valuable insight into the suite of BC measurements throughout the Arctic, and perspective on previous model comparison with only one instrument type. EBC concentrations measured by the Aethalometer are biased high by a factor of 2 relative to rBC measurements, due to the

presence of absorbing substances other than BC (e.g. brown carbon and mineral dust), extinction issues associated with the filter matrix and uncertainties in MAC values (Sharma et al., submitted). EC concentrations are lower than EBC concentrations from the Aethalometer, yet still high relative to rBC. PSAP EBC concentrations are close to the average of EC and rBC concentrations throughout the year. At Barrow, EBC

concentrations from the Aetholometer are higher than those from the PSAP, especially in summer when the Aetholometer shows a pronounced increase in concentrations to around 55 ng m$^{-3}$, whereas PSAP measurements reach a minimum for the year of 10 ng m$^{-3}$. This contrary behavior could be due to the unintentional exclusion of biomass burning plumes in the local pollution data screening performed for PSAP measurements

at Barrow (Stohl et al., 2006). Higher EBC concentrations measured by the Aetholometer compared with the PSAP are also observed at Ny-Ålesund.

Following Sharma et al. (submitted), we treat the best estimate of measured BC surface concentrations at Alert as the average of rBC and EC measurements, as shown by the thick black line with squares in Fig. 3. Since the PSAP EBC concentrations are close to the

average of rBC and EC measurements throughout the year at Alert, we adopt the PSAP EBC measurements as the best estimate of surface BC at Barrow and Ny-Ålesund. The seasonal variations of surface BC at the three sites show similar features, characterized by higher concentrations in winter and early spring than in summer. At Ny-Ålesund, peak months are March and April, slightly later than at the other sites (January and

February). BC concentrations at Ny-Ålesund are generally lower than those at the other sites.

The surface BC concentrations from measurements are used to constrain emissions in simulations. Table 1 summarizes the RMSE between measurements and different





simulations. The green line in Fig. 3 shows simulated surface BC concentrations using anthropogenic emissions of BC from the Bond et al. (2007) non-seasonal emission inventory. Stohl et al. (2013) found that accounting for emissions of BC from gas flaring and from seasonal variation of residential heating improved their simulation with a

particle dispersion model (FLEXPART) during winter and early spring. Our simulation at Alert and Barrow in winter and spring is also improved by using the HTAP emissions that include seasonal variation of residential heating, as shown by comparing the yellow and the blue lines. Adding flaring emissions to the HTAP inventory further improves the consistency with measurements in winter and early spring at Alert and Barrow by

decreasing the bias by about a factor of 2 and reducing the rRMSE to 5.6% at Alert and 13% at Barrow. At Barrow all simulations show a distinct peak in July, which is due to the timing of biomass burning. Eckhardt et al. (2015) similarly observed enhanced concentrations in July at Barrow in four models (FLEXPART, DEHM, CESM1-CAM5 and ECHAM6-HAM2) driven with the GFED3 inventory for biomass burning emissions. At Ny-

Ålesund, the effects of different emissions on BC concentrations are comparable to those at Alert and Barrow, yet all simulations overestimate measured concentrations for most of the year, potentially indicating insufficient wet deposition from riming in mixed phase clouds that occurs more frequently at this site (Qi et al., 2017b).

Figure 4 shows vertical profiles of BC concentrations at Alert and Ny-Ålesund averaged

from the NETCARE 2015, the PAMARCMiP 2009 and the PAMARCMiP 2011 campaigns, along with the best estimate of ground-based measurements of April BC concentrations averaged over 2009 and 2011. Barrow is not included here due to limited number of airborne measurements (a total of 12 measurements at all pressures). The measured profile at Alert exhibits layered structure with roughly constant concentrations near the

surface and enhanced concentrations in the middle troposphere that are attributable to a plume on April $8^{th}$ 2015 around 660-760hPa with a peak concentration of 128 ng m$^{-3}$. The mean ground-based measurements of BC concentrations at Alert are higher than airborne measurements at the same pressure by ~10 ng m$^{-3}$. Including only rBC measurements in ground-based mean concentrations reduces the difference with




airborne rBC measurements to less than 5 ng m$^{-3}$. At Ny-Ålesund, the measured vertical profile exhibits a zigzag shape that arises from averaging multiple years each with individual features. The mean April ground-based concentration (20 ng m$^{-3}$) is about half that of the airborne measurements (37 ng m$^{-3}$) at the same pressure.

Figure 5 shows spring vertical distributions of BC averaged over all points along the flight tracks of the three campaigns in Fig. 1 for measurements and simulations. The measured rBC concentrations remain roughly constant (~38 ng m$^{-3}$) from the surface to 700hPa, followed by an enhancement to around 50 ng m$^{-3}$ between 700hPa – 500hPa, and then a rapid decrease with altitude. This vertical distribution is similar to the

measurements of the ARCTAS aircraft campaign in the Arctic in spring 2008 (Wang et al., 2011), though the magnitude of concentrations in this work is lower by a factor of about 2, likely because the Arctic was substantially influenced by strong biomass burning in northern Eurasia during the ARCTAS in spring 2008 (Warneke et al., 2009).

All simulations generally represent the relative vertical distribution of BC from

measurements. They all show that BC concentrations gradually increase from the surface to 700hPa, and then decrease with altitude above 500hPa, yet none represent the enhancement between 700-500hPa that is also an issue in many other models (e.g. GISS, CAM; Koch et al., 2009). Despite the comparable distributions, the magnitudes of concentrations simulated with different emissions vary substantially. Their consistencies

with airborne measurements are summarized in Table 1.

Fig. 5 shows that the apparent bias of 40% rRMSE (17 ng m$^{-3}$ RMSE) in simulated concentrations with the Bond et al. (2007) non-seasonal inventory is reduced to 27% rRMSE (11 ng m$^{-3}$ RMSE) by the HTAP inventory with non-seasonal residential heating. The improvement is larger aloft than near-surface, indicating that the increased BC

emissions in Asia in the HTAP inventory (discussed in Section 2) that tends to contribute to Arctic aerosols in the middle troposphere (Klonecki, 2003) substantially contributes to the improvement. The bias versus measurements is further reduced to 23% rRMSE (9.4 ng m$^{-3}$ RMSE) by the HTAP emissions with seasonal residential heating, with larger





improvement below 600hPa. Adding flaring emissions further improves the consistency (17% rRMSE; 7.2 ng m$^{-3}$ RMSE) with measurements at all levels with larger effects in the lower troposphere, especially near the surface where RMSE is only 3.2 ng m$^{-3}$. The substantial portion (93%) of flaring in BC emissions within the Arctic (Fig. 2) explains the

larger effect near the ground. The remaining underestimation of 14 ng m$^{-3}$ RMSE in 500-700hPa in the HTAP+flaring simulation is perhaps due to missing plumes including the one on April 8$^{th}$ 2015 as discussed in Fig. 4 and other plumes near Barrow, Alert and Ny-Ålesund as will be discussed below.

Figure 6 (top-left and top-middle) shows the spatial distribution of BC concentrations

from aircraft measurements gridded onto the GEOS-Chem grid along with that from the HTAP+flaring simulation. The simulation represents well the spatial distribution of BC measurements, with concentrations of 30-70 ng m$^{-3}$ near Barrow and Ny-Ålesund and lower concentrations of 20-40 ng m$^{-3}$ near Alert, yet the simulation underestimates concentrations at three hotspots (labeled as A, B, C). Hotspot A is near Barrow along the

coast of the Beaufort Sea that is affected by a plume around 680 hPa (average concentration ~125 ng m$^{-3}$) on April 6$^{th}$ 2011 and a plume around 580 hPa (average concentration ~ 162 ng m$^{-3}$) on April 20$^{th}$ 2015. Hotspot B is west of the Baffin Bay in Nunavut that is affected by a plume near 700hPa (average concentration ~131 ng m$^{-3}$) on April 10$^{th}$ 2011. These plumes appear to originate mostly from eastern and southern

Asia as revealed by sensitivity simulations. Hotspot C is near Ny-Ålesund with an average concentration of 100 ng m$^{-3}$ that is caused by a plume around 670 hPa on May 5th 2011 with a likely origin from Eurasia. The simulation well represents the spatial distribution of the plumes near Ny-Ålesund and in Nunavut, despite an underestimation in magnitude, but misses the plume near Barrow. The misrepresentation of these plumes

in the simulation may explain the significant underestimation of BC concentrations between 500-700hPa in Fig. 5. The top-right panel of Fig. 6 shows mean simulated BC concentrations between 500-700 hPa in April. Concentrations are highest (~70 ng m$^{-3}$) in northeastern Russia and near Barrow, with a gradual decrease eastward to around 50 ng m$^{-3}$ near Alert to reach the lowest concentrations of below 40 ng m$^{-3}$ in the southern



Arctic near Ny-Ålesund. This gradient illustrates the overall sources and transport pathways affecting BC in the Arctic middle troposphere in springtime. The next section will investigate that the enhanced concentrations in northeastern Russia and their relation to sources in eastern and southern Asia.

Figure 6 (bottom row) shows pan-Arctic spatial distributions of BC column (1000 hPa – 300 hPa) concentrations from the HTAP+flaring simulation for January, April and July. April has the highest column concentrations with higher overall concentrations in the eastern Arctic (particularly northeastern and northwestern Russia) than in the western Arctic, suggesting the large influence of northern Asia and Europe on Arctic aerosol

loadings in spring. January has the lowest overall column concentrations, reflecting inhibited transport into the Arctic due to the southward extension of the Arctic front in winter (Stohl, 2006). Nevertheless, concentrations near northern Russia remain high, influenced by large flaring emissions there (Fig. 2) as discussed further below. In July, North America exhibits remarkably high concentrations from biomass burning as will be

discussed further in section 3.2.

Since BC concentrations simulated with HTAP+flaring exhibit overall consistency with the measured seasonal variation, and the measured vertical and spatial distributions, we use this inventory in the following simulations for source attributions.

3.2 Source attribution of BC in the Arctic

Figure 7 (left) shows the contribution of anthropogenic emissions from regions defined in Fig. 1, and of biomass burning from North America and the rest of the world, to springtime airborne BC along the flight tracks of the three aircraft campaigns in Fig. 2. The contributions are quantified by excluding the regional emissions there. At all levels, anthropogenic emissions explain more than 90% of BC concentrations, of which 56% is

contributed by eastern and southern Asia, followed by Europe with a contribution of 19%. Biomass burning is minor (~8%) compared to anthropogenic emissions in the contribution to springtime Arctic BC loadings, and the biomass burning impact on the springtime Arctic almost exclusively originates from regions other than North America.





Qi et al. (2017a) found a substantial contribution from Siberian biomass burning to the Arctic lower troposphere in April 2008. The relative contribution of anthropogenic emissions from each source region varies with altitude, reflecting different transport pathways. The influence of eastern and southern Asia increases considerably with

altitude, with a contribution of 66% between 400 - 700 hPa and 46% between 900 - 1000 hPa, because transport from mid-latitudes follows isentropic surfaces that slope upward toward the middle or upper troposphere in the Arctic (Klonecki et al. 2003). In contrast, the influence of northern Asia decreases rapidly with altitude by a factor of 4 from the surface to 700hPa, reflecting transport from sufficiently cold regions along the

low-level isentropic surfaces into the Arctic and direct transport within the polar dome (Klonecki et al., 2003; Stohl, 2006). The impact of Europe is roughly uniform throughout the troposphere, suggesting both of the above pathways are possible.

The gas flaring contribution to the springtime vertical BC concentration is shown as the red shading in Fig. 5. The contribution decreases with altitude from ~20% near the

surface to <10% above 800 hPa because flaring occurs almost exclusively below 2 km a.s.l (Stohl et al., 2013) and because the high-latitude sources of flaring limit isentropic lifting in the polar dome (Stohl, 2006).

Figure 7 (right) shows the annual mean vertical contribution of anthropogenic emissions from each source region and of biomass burning to Arctic BC. Anthropogenic emissions

from eastern and southern Asia (37%) and biomass burning emissions (25%) are major sources of Arctic tropospheric BC, along with a substantial contribution (43%) from anthropogenic emissions in northern Asia near the surface (>900hPa). Unlike in spring, roughly half of biomass burning BC originates from North America in the annual attribution, which reflects that North America biomass burning is often later in the year.

Compared to springtime, the annual anthropogenic contribution from eastern and southern Asia is smaller and that from northern Asia is substantially larger in the lower troposphere. This suggests that long-range transport from eastern and southern Asia is more favorable in spring, and that proximal transport from northern Asia is more





efficient in winter. Liu et al. (2015) found that the efficient springtime transport from eastern and southern Asian was facilitated by warm conveyor belts through which air parcels were rapidly uplifted on timescales of 1–2 days followed by long-range transport. Stohl (2006) found that low-level transport from northern Asia was efficient
in winter, when strong diabatic cooling in northern Asia extended the Arctic front to the south of northern Asian sources, facilitating their influence on the Arctic at low altitudes.

Previous studies found generally similar source attributions of vertical BC in the Arctic. Liu et al. (2015) found that the contribution from anthropogenic emissions in eastern
and southern Asia increased substantially in the middle troposphere, and that the contributions from biomass burning and from North American anthropogenic emissions are minor (<10%) throughout the troposphere for March 2013. However, the larger effect of European anthropogenic emissions that they find is likely due to their focus on the European Arctic. Shindell et al. (2008) studied the sensitivity of annual average
Arctic BC concentrations to emissions and found that East Asian anthropogenic emissions had larger contributions at 500hPa (38%) than near the surface (13%; 900 hPa -1000 hPa), similar to our results. Sharma et al. (2013) examined simulations for 1990-2005 and suggested that in winter the anthropogenic contribution from East Asia on Arctic BC burden is 3 fold larger than that on surface BC, consistent with our results.
However, they found that Europe and the former Soviet Union explained more than one half of Arctic BC burden, whereas we find eastern and southern Asia is the largest contributor to BC burden. The difference likely arises from trends in anthropogenic emissions with reductions from Europe and increases in eastern and southern Asia as discussed further below.

Figure 8 shows the simulated source attribution of surface BC at Alert, Barrow and Ny-Ålesund. For all stations, anthropogenic emissions from northern Asia, eastern and southern Asia, and Europe are major contributors to high concentrations of BC in winter and early spring. In summer, anthropogenic contributions decline rapidly while biomass




burning predominantly from North America becomes the primary source. Anthropogenic emissions from North America have a minor influence (<12%) throughout the year at all stations. The relative contributions from each source of BC are similar at Alert and Barrow in winter and spring, with the largest contributions from

anthropogenic emission in northern Asia in winter (~50%), and in eastern and southern Asia in spring (~40%). Barrow shows a pronounced peak in summer, more than 90% of which is explained by biomass burning from North America. At Ny-Ålesund, anthropogenic emissions in Europe and northern Asia are significant sources of BC in winter and early spring with a contribution of ~30% from each source.

The contributions from gas flaring to surface BC concentrations are shown as the red shadings in Fig. 3. Flaring accounts for ~25% of concentrations in winter and spring and less than 5% in summer at all stations except Ny-Ålesund where flaring contributes 14% of BC in summer. This result is consistent with Stohl et al. (2013) who studied the flaring contribution to surface BC concentrations at Arctic stations using the FLEXPART model.

We also investigate the influence of international shipping from the HTAP v2 inventory for 2010 on Arctic surface BC concentrations, and found the contribution is less than 1% at all stations owing to the small magnitude of emissions (<1% of total anthropogenic BC emissions globally and within the Arctic). This source is expected to increase substantially over the coming decades (Winther et al., 2014).

Our source attribution of Arctic surface BC is consistent with that of Koch and Hansen (2005) who investigated the origins of Arctic BC using a general circulation model and found that Russia, Europe and south Asia each accounted for 20% - 30% of springtime surface BC . However, some studies (e.g. Gong et al., 2010; Sharma et al., 2013; Shindell et al., 2008; Stohl, 2006) suggested lower contributions (<10%) from eastern and

southern Asia to Arctic surface BC than our results (~25% for Alert and Barrow; 16% for Ny-Ålesund annually). The main difference is due to emission trends that our anthropogenic emissions from eastern and southern Asia are generally 30% higher than those in other studies (e.g. Sharma et al., 2013; Shindell et al., 2008) and that





anthropogenic emissions in Europe are halved in this work. Stohl et al. (2013) also found

the distinct summer peaks from biomass burning at Arctic stations as in our study. The

weak influence of North American anthropogenic emission on the Arctic surface BC

concentrations was found in previous studies (Brock et al., 2011; Liu et al., 2015). The

likely reason is that the majority of anthropogenic emissions in North America take

place at lower latitudes with relatively high potential temperature that is inhibited from

reaching the Arctic, and that the predominant transport from North America to the

Arctic is over the warm Atlantic where air masses experience strong diabatic heating

due to the instability in the atmosphere and heavy precipitation in the storm track

(Klonecki, 2003). Jiao and Flanner (2016) also suggested that geostrophic winds inhibit

pole-ward transport of North American emissions.

Figure 9 shows the contributions to Arctic BC column concentrations from changes in

local emissions in 2011 as calculated with the GEOS-Chem adjoint. Pronounced seasonal

variation and spatial heterogeneity are found. Sources in January are strongly influenced

by specific Asian regions including western Siberia, eastern China and the Indo-Gangetic

Plain, whereas sources in other seasons are more widespread across Europe and North

America. Several hotspots are found in each season. In January, oilfields in western

Siberia have a total impact of 13% on Arctic BC loadings, of which 4.4% is from the

Timan-Pechora basin oilfield and 6.4% from the West Siberia oilfields, suggesting that

the influence of western Siberia is comparable to the total influence of continental

Europe and North America (~6.5% each in January). Considerable flaring emissions (67%

of total flaring emissions north of 60°N in January) and close proximity to the Arctic

contribute to the substantial influence of those oilfields in western Siberia. The Indo-

Gangetic Plain also exhibits considerable impact (7.2%) to the Arctic, reflecting the

substantial emissions there as shown in Fig. 1. In April, the influence of western Siberia

decreases to 4.4% as the northward retreat of the Arctic front eliminates the direct

transport from western Siberia to the Arctic within the polar dome (Stohl, 2006). In

contrast, contributions from emissions in eastern China (25%) and North America (8.2%)

are enhanced when the warming of the surface leads to higher potential temperature




over the Arctic that facilitates transport of air masses from warm regions (e.g. the US and Asia) to the Arctic (Klonecki, 2003). Emission contributions to Arctic BC loadings are generally weak in July, but the Tarim oilfield in western China stands out as the second most influential (3.2%) grid cell to the Arctic, which is comparable to a half of the impact

of continental Europe (6%). The Tarim oilfield is located in a high altitude (~1000 m) arid region (Taklamakan Desert). Considerable flaring emissions, less efficient wet scavenging and elevation all facilitate its large contribution to the Arctic. The contribution from North America is the largest (13%) in July, consistent with the remarkably high BC loadings over high-latitude North America as shown in Fig. 6

(bottom right). Annually, eastern China (15%), western Siberia (6.5 %) and the Indo-Gangetic Plain (6.3%) have the largest impact on Arctic BC loadings, along with a noteworthy contribution from the Tarim oilfield (2.6%). At continental scales, eastern and southern Asia contribute 40% to the Arctic BC loadings. Northern Asia, North America and Europe each make a contribution of ~10%, consistent with the vertical

source attribution from sensitivity simulations in Fig. 7 (right). BC emissions within the Arctic generally contribute less than 3% of Arctic BC loadings in all seasons except for January (5%).

4. Conclusions

Airborne measurements of BC concentrations taken across the Arctic during the

NETCARE 2015, the PAMARCMiP 2009 and the PAMARCMiP 2011 campaigns, along with long-term ground-based measurements of BC concentrations from three Arctic stations (Alert, Barrow and Ny-Ålesund) were interpreted with the GEOS-Chem chemical transport model and its adjoint to quantify the sources of Arctic BC. Measurements from multiple BC instruments (rBC, EC, EBC) were examined to quantify Arctic BC

concentrations. We relied on rBC and EC measurements, and on EBC inferred from PSAP absorption measurements with a MAC calibrated to rBC and EC measurements. Our simulations with the addition of gas flaring emissions were consistent with ground-based measurements of BC concentrations at Alert and Barrow in winter and spring



(rRMSE < 13%), and with airborne measurements of BC vertical profile across the Arctic (rRMSE=17%). However, our simulation underrepresented an enhancement of BC concentrations between 500-700hPa that was affected by several plumes near Alert, Barrow and Ny-Ålesund with an origin from Eurasia.

Sensitivity simulations with the GEOS-Chem model were conducted to assess the contribution of geographic sources to Arctic BC. The springtime Arctic tropospheric BC burden are predominantly affected by anthropogenic emissions from eastern and southern Asia (56% from 1013 hPa to 400 hPa) with larger contributions aloft (66% between 400-700 hPa) than near the surface (46% below 900 hPa), reflecting long-range

transport in the middle troposphere. Anthropogenic emissions from northern Asia had considerable contributions in the lower troposphere (27% below 900hPa) in spring due to low-level proximal transport. Annually, biomass burning (25%) and anthropogenic emissions from eastern and southern Asia (37%) were major sources of Arctic tropospheric BC. Northern Asian anthropogenic emissions were the largest contributor

(43%) near the surface (>900hPa). Surface BC was largely influenced by anthropogenic emissions from northern Asia (>50%) in winter and eastern and southern Asia in spring (~40%) at both Alert and Barrow, and from Europe (~30%) and northern Asia (~30%) at Ny-Ålesund in winter and early spring. Biomass burning primarily from North America, was the most important contributor to surface BC at all stations in summer, especially at

Barrow where North American biomass burning dominated BC in July and August. Anthropogenic emissions in North America had minor influence (<10%) on both surface and vertical BC concentrations, since anthropogenic emissions in North America are primarily at lower latitudes with inhibited transport into the Arctic.

Our adjoint simulations indicated pronounced spatial and seasonal heterogeneity in the

contribution of emissions to Arctic BC column concentrations to emissions. Eastern China (15%) and western Siberia (6.5%) had a noteworthy influence on Arctic BC loadings on an annual average. The Tarim oilfield stood out as the second most influential grid cell with an annual contribution of 2.6%. Gas flaring emissions from





oilfields in western Siberia had a striking impact (13%) on the Arctic in January, which was comparable to the total impact of continental Europe and North America (6.5% each in January). The Indo-Gangetic Plain also exhibited moderate influence (7.2%) on the Arctic in January.

Further work to reconcile the different BC mass concentrations measured by different instruments would be valuable to reduce uncertainties in BC studies not only in the Arctic but also globally.

Acknowledgement

The authors acknowledge the financial support provided for NETCARE through the

Climate Change and Atmospheric Research Program at NSERC Canada. We also acknowledge the World Data Centre for Aerosol, in which BC measurements from Arctic stations are hosted (http://ebas.nilu.no). We thank all operators at Barrow and Ny-Ålesund stations for maintaining and providing ground-based BC measurements. We also thank the developers of the HTAP and ECLIPSE emission inventories.

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

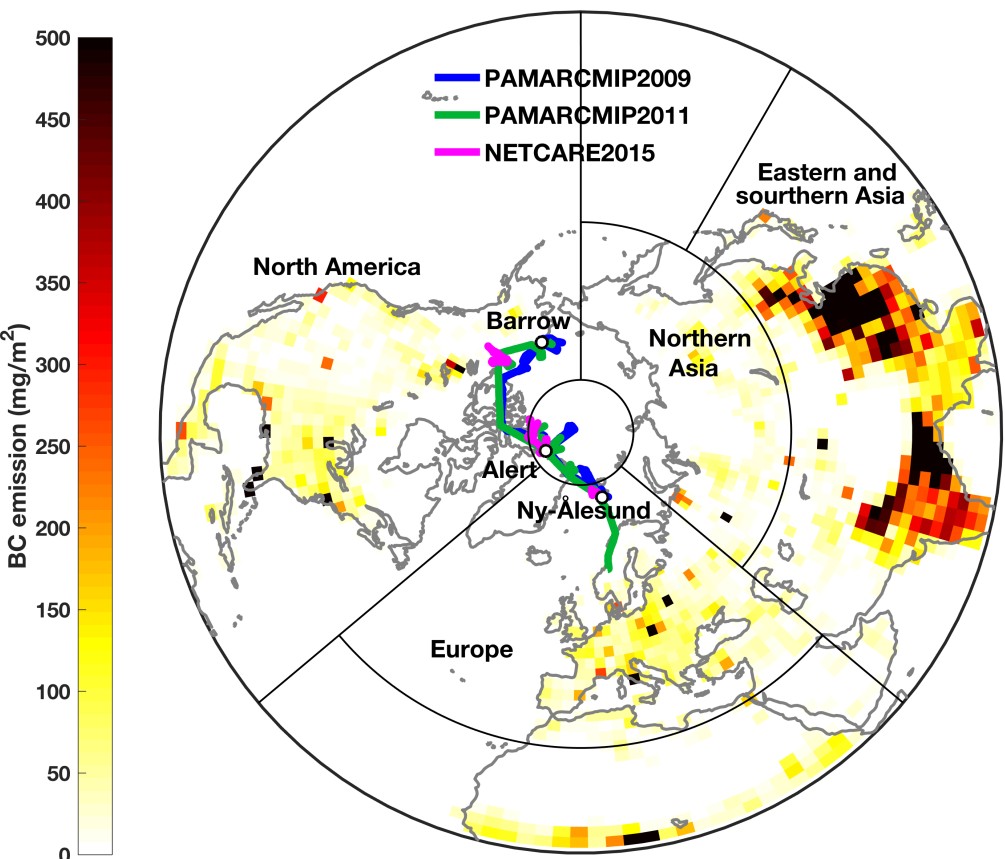

Figure 1. The colormap indicates annual BC emissions averaged over 2009, 2011 and
2015 as used in the GEOS-Chem simulation. Black open circles indicate the locations of
ground monitoring stations (Alert, Barrow and Ny-Ålesund). Colored lines indicate the
10    flight tracks of the NETCARE 2015 (April 5th-21st), the PAMARCMiP 2009 (April 1st -25th)



and the PAMARCMiP 2011 (Mar 30th – May 5th) campaigns. Black lines outline the

source regions used in this study.

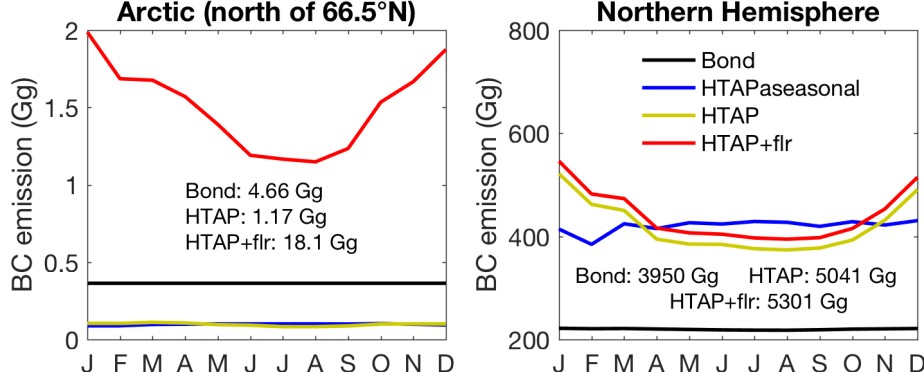

Figure 2. Anthropogenic BC emissions. Lines indicate monthly anthropogenic BC

emissions from the Bond et al. (2007) non-seasonal inventory for 2000, the HTAP

inventory for 2010, the HTAP inventory with non-seasonal emissions from residential

heating, and the HTAP with additional flaring emissions for 2010. Annual values are

10      given in the text.





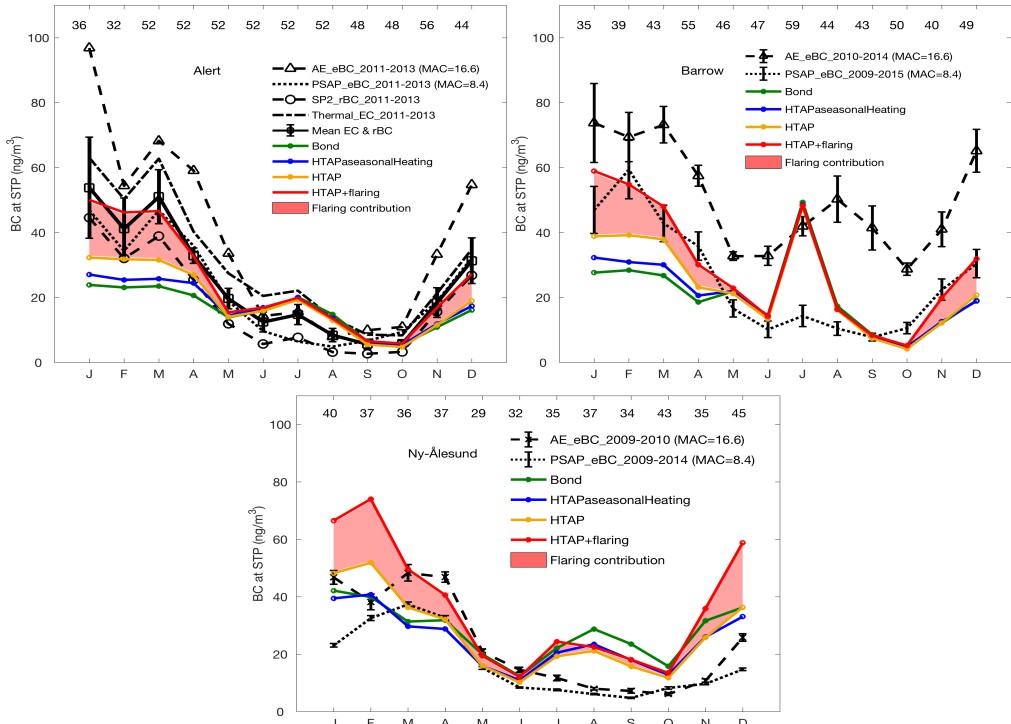

Figure 3. Seasonal variation of surface BC concentrations from measurements and simulations at selected Arctic stations. Black lines represent measurements from different instruments according to the legend. Error bars represent standard errors. The

5    thick black line with squares at Alert is the average of rBC and EC concentrations. Error bars on the thick black line denote standard errors of monthly mean BC concentrations across instruments that are included in the calculation. Red shadings are the contributions from flaring to BC concentrations. Numbers below the top x-axis denote the total number of weekly observations from all available instruments in each month.

10   Simulated monthly BC concentrations are the monthly averages of simulated concentrations for 2009, 2011 and 2015. Simulations use different emission inventories that are represented in color according to the legend. Concentrations from measurements and simulations are all calculated at standard temperature and pressure (STP).





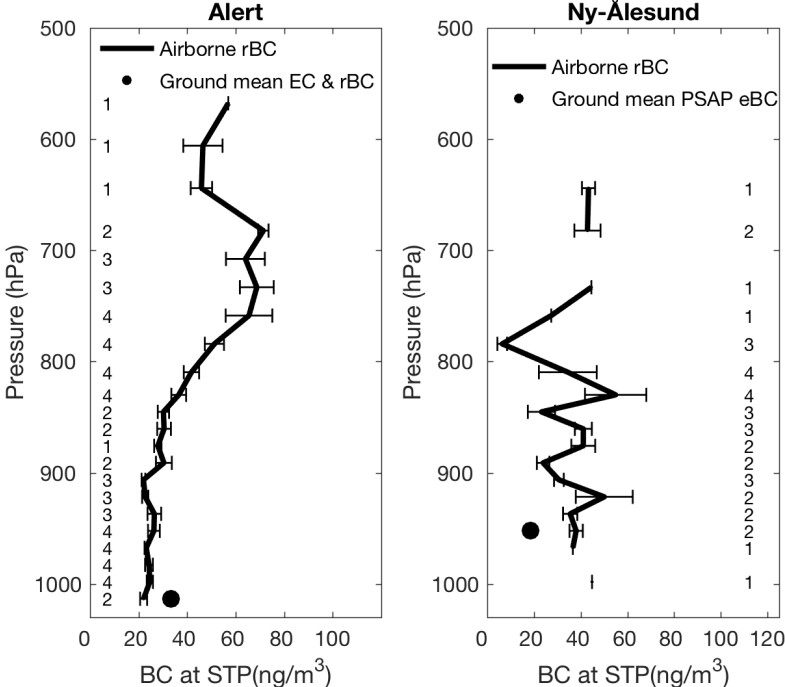

Figure 4. Vertical profile of BC concentrations averaged from all points along the flight
tracks of the three aircraft campaigns (NETCARE 2015, the PAMARCMiP 2009 and the
PAMARCMiP 2011) in Alert and Ny-Ålesund areas, along with the best estimate of April
5    BC concentrations from ground-based measurements averaged for 2009 and 2011. The
Alert area is defined as 59°W-65°W, 81.3°N-83.4°N and the Ny-Ålesund area is within
12°E-18°E, 77.8°N-79.1°N. Numbers along the y-axis are the number of airborne
measurements in each pressure bin. All concentrations are presented at STP.



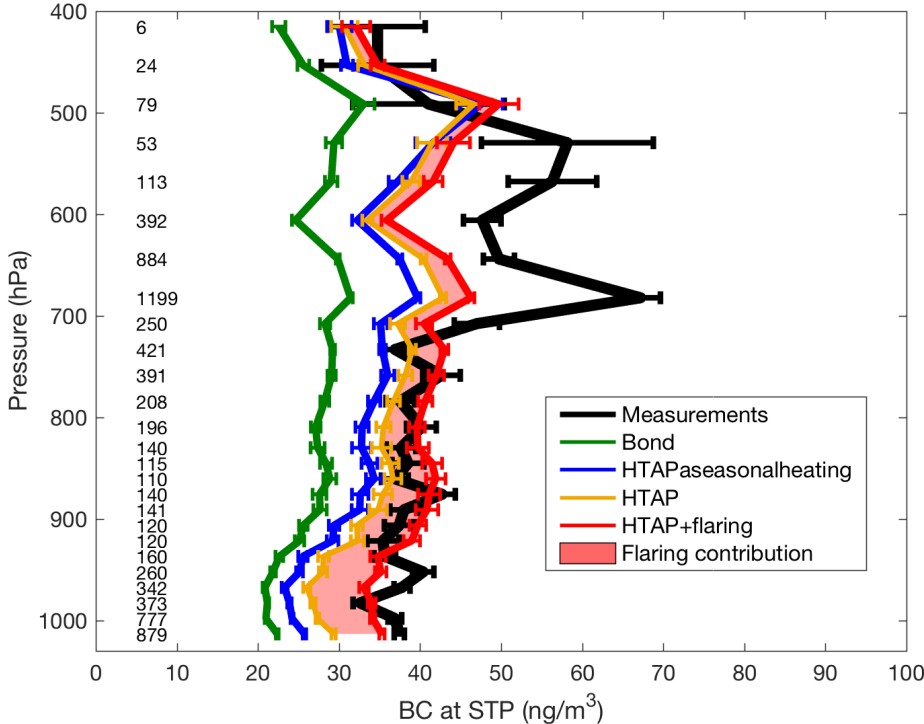

Figure 5. Mean spring vertical profiles of BC concentrations from measurements and simulations averaged over 50 hPa pressure bins from all points along the flight tracks of the NETCARE 2015, the PAMARCMiP 2009 and the PAMARCMiP 2011 campaigns. The red shading denotes the contribution of flaring to BC concentrations. Simulated vertical profiles of BC are coincidently sampled with airborne measurements and are averaged to the GEOS-Chem vertical resolution. Simulations include different emission inventories that are represented in different lines according to the legend. Error bars are standard errors. Numbers along the y-axis represent the number of measurements in each pressure bin. All concentrations are presented at STP.




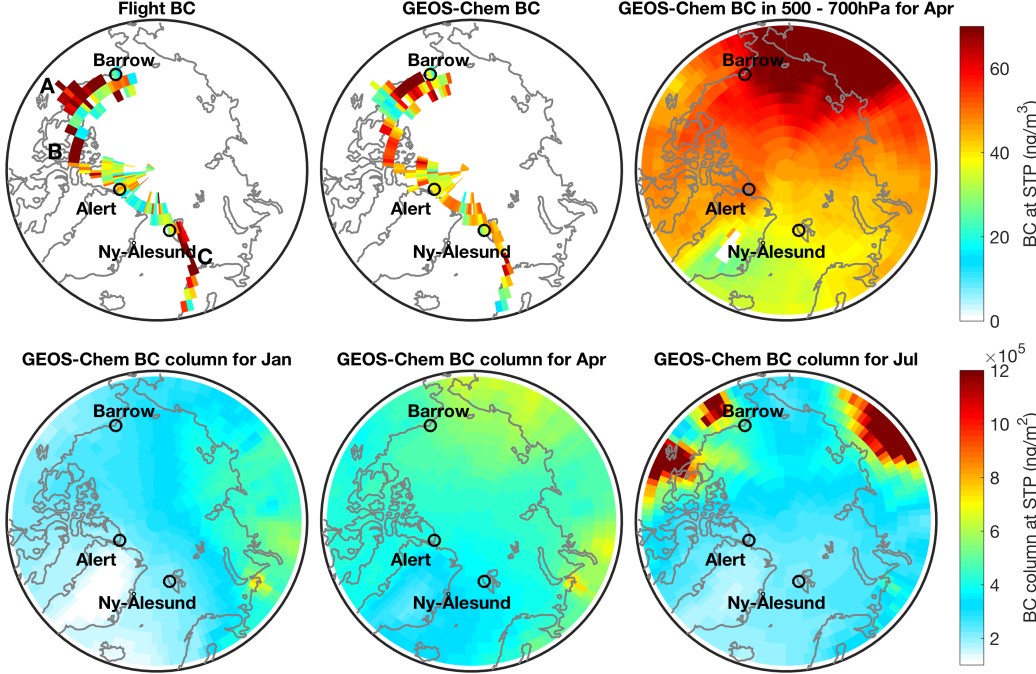

Figure 6. Top left: BC concentrations from the NETCARE 2015, PAMARCMiP 2009 and 2011 aircraft campaigns averaged on the GEOS-Chem grid, along with three hotspots labeled as A, B, C. Top middle: BC concentrations from GEOS-Chem simulations coincidently sampled with flight measurements. Top right: BC concentrations between 500-700 hPa simulated with the HTAP+flaring emissions in April averaged over 2009, 2011 and 2015. Circles are ground monitoring stations. Bottom: pan-Arctic BC column concentrations simulated with the HTAP+flaring emissions for January (left), April (middle) and July (right) averaged over 2009, 2011 and 2015. All concentrations are at STP.





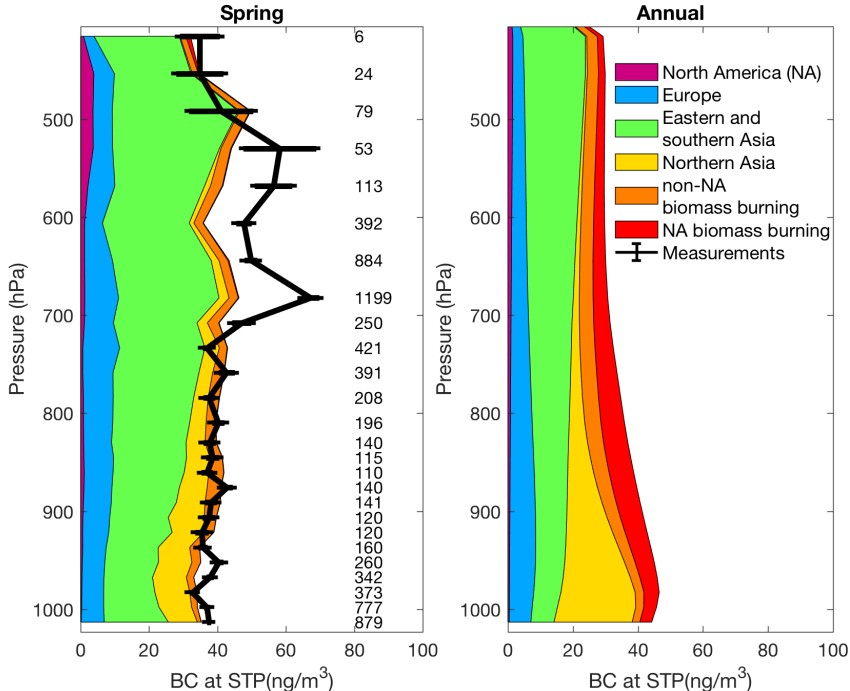

Figure 7. Left: Mean spring BC vertical profiles from flight measurements and simulations that are color-coded to anthropogenic sources from regions defined in Fig 1. and biomass burning sources from North America and the rest of the world. Flight measurements and error bars are the same as in Fig. 5. Simulated vertical profiles of BC are taken coincidently with flight measurements. Numbers along the y-axis represent the number of measurements in each pressure bin. Right: annual mean vertical profile of BC for the entire Arctic from simulations that are color-coded to source regions. Concentrations are all presented at STP.





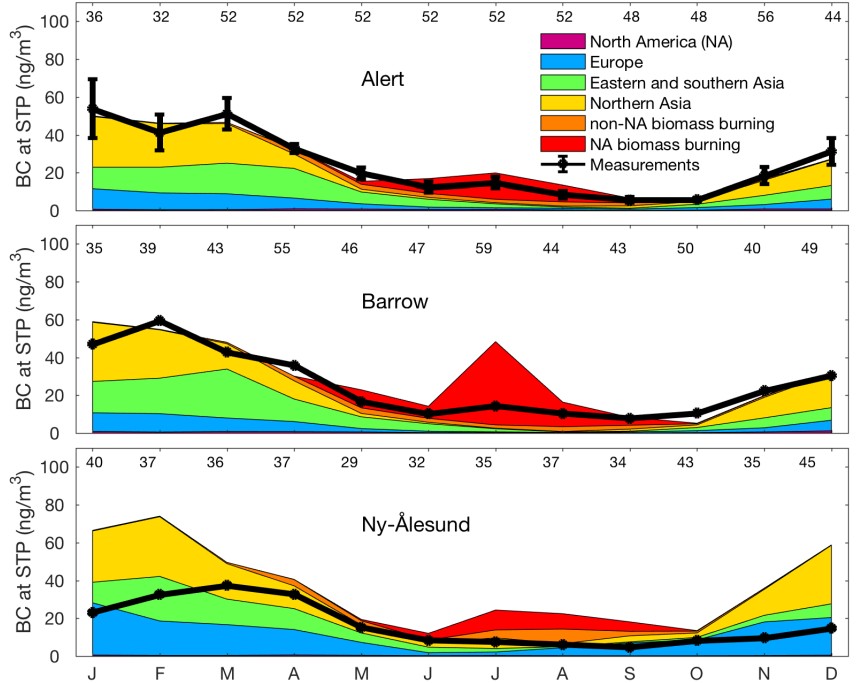

Figure 8. Monthly variation of BC surface concentrations at selected Arctic stations from measurements and simulations that are color-coded to anthropogenic sources from regions defined in Fig 1. and biomass burning sources from North America and the rest of the world. The measured monthly mean concentrations of BC and error bars are the same as the best estimate of surface BC concentrations in Fig. 3. Simulated monthly concentrations are monthly averages of 2009, 2011 and 2015. Numbers below the top x-axis denote the total number of weekly observations from all available instruments in each month. Concentrations are all presented at STP.





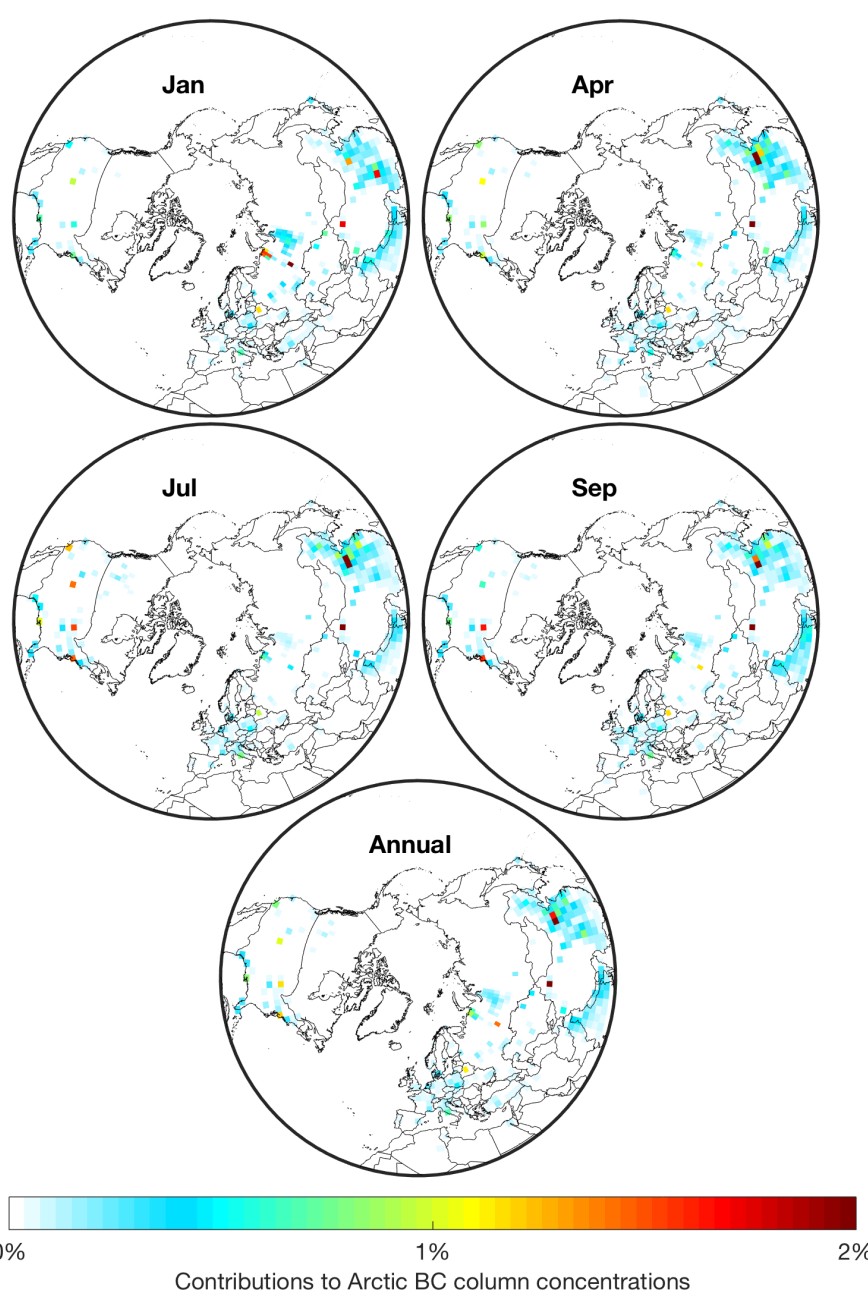

Contributions to Arctic BC column concentrations

Figure 9. Contributions to Arctic BC column concentrations from changes in local

emissions (as percent change in Arctic BC column concentration per fractional change in

emissions) in 2011. Local emissions include anthropogenic and biomass burning





emissions. The annual map is the average of contributions in January, April, July and
September.

5    Table 1. Summary of root mean square error (RMSE) between simulations with different
emissions and measurements for BC surface concentrations at Arctic stations (in
reference to Fig. 3) and for vertical concentrations from airborne measurements (in
reference to Fig. 5).

| RMSE(ng m$^{-3}$) | Alert | Barrow | Ny-Ålesund | Vertical |
|---|---|---|---|---|
| Bond | 13 | 17 | 15 | 17 |
| HTAPaseasonalheating | 11 | 16 | 12 | 11 |
| HTAPheating | 8.7 | 13 | 14 | 9.4 |
| HTAPheatingflaring | 3.7 | 11 | 25 | 7.2 |