# Peer review of "Source attribution of Arctic black carbon constrained by aircraft and surface measurements"

_Atmospheric Chemistry and Physics, 2017_

## Referee Comment (RC1) · Anonymous Referee #1 · 9 May 2017

General comments : This work investigates the source attribution of Arctic black caron using a global CTM and surface (at 3 stations) and airborne (NETCARE, PAMARCMiP) observations. Using different simulations switching off some of the emission sources (e.g. flares, seasonal heating) as well as a global CTM adjoint, the authors determine the spatially resolved source contributions to receptor locations. They highlight the influence of different pollution sources and the resulting vertical and spatial distribution of BC. This study is interesting, scientifically important and sound for ACP. My main comment is that the authors oversell a bit the performance of the model to reproduce the vertical gradient observed in the BC concentrations. The model is actually doing a reasonable job of capturing the general features of the spatial BC distribution, though not doing a perfect job of simulating the concentrations. Most comments listed below are minor clarifications. Once these points are addressed satisfactory, the paper should

in my opinion be suitable for publication in ACP.

—

Specific comments :

P1L15 : I found the abstract too long, written like a conclusion section. Please try to make it more concise and underline the key messages from your study.

P1L27-28 : I think you oversell the performance of the model in reproducing the vertical profile of BC. I don't think you can say that the simulations are "consistent" with the airborne measurements of BC. See for example Fig. 5 where the simulated profiles are flat, especially between 700 and 500 hPa, whereas the observations demonstrate a significant enhancement of BC concentrations. See more comments below (P15L14, P17L17 or P23L1).

P6L4-7 : The way it is phrased makes it difficult to understand what is new in this study. Do the authors develop the approach ? Do they apply it to a different/longer period than that in Qi et al. (2017a) ? Highlight the differences between the methodologies detailed in Kopacz et al. (2011) or Qi et al. (2017a).

P6L8 : Say that GEOS-Chem is a global CTM. It was only mentioned in the abstract.

P7L5-6 : Can you justify why no scattering corrections to the aethalometer measurements contrary to what is done for the PSAP data (P7L12) ?

P10L11-16 : How are fire emissions injected in the model ? Are the emissions confined to the surface or does the model takes into account the variability in injection heights (function of burnt vegetation, intensity of fires, buoyancy, ...) ?

P10L22 : The description of the simulation is not given in this study. The authors give only a reference to an "old" paper (Park et al., 2003), which cannot be taken as a good reference to fully decribe what has been really done in this work. Please give additional details : period of the simulation, horizontal and vertical resolutions, . . . A

lot of modifications have been included in the GEOS-Chem model from the study of Park et al. (2003). Some of them are likely to produce substantial changes in the BC distributions (e.g. wet deposition). Try to underline the main improvements.

P10L26 : Does the model include wet deposition also in subgrid convective updrafts?

P13L8 : The authors explain that EBC concentrations are biased high relative to rBC measurements because of absorbing components and errors in the coefficient used to derive BC concentrations from absorption coefficients. But the fact that EC concentrations are always larger than rBC is never explained. Is it similar to what was observed in other studies ? Can it be caused by a difference in sampling the plumes ?

P13L14-16 : May this discrepancy between aethalometer and PSAP measurements be also ascribed to differences in the calibration method : scattering corrections applied to PSAP but not to aethalometer data ? And/or spectral dependency of the absorption coefficient of other substances than BC ?

P14L11-14 : The peak observed in model simulations at Barrow is explained by a wrong timing of biomass burning emissions. Shouldn't it be the case for all models compared in the Eckhardt et al. (2015) paper if they all use the same fire emissions (with the same timing) ? What is the role of injection heights in summer ? And what is the contribution of wet deposition processes in the different models ?

P15L14 : I think this sentence oversells the model prediction. The main feature noticed in the observed BC profile is the significant enhancement between 700 and 500 hPa, which is not captured at all by the simulations. So the model does not "generally represent " the relative vertical distribution of BC. The next sentence in the manuscript is a much better way to say this.

P15L17-18 : The way the sentence has been written gives the impression that most models also strongly underestimate BC in the mid or upper-troposphere. You cannot say this without also refering to more recent studies, highlighting either an overestimation aloft (e.g. Sharma et al. 2013 ; Wang et al., 2014), a improvement in the mid troposphere (e.g. Breider al., 2014 ; Raut et al., 2017) or a very distinct behaviour between models (as in Eckhardt et al., 2015). In particular both Breider al. (2014) and Wang et al. (2014) used the GEOS-Chem model to represent BC in the Arctic and the results are not similar. This latter study suggested an incorrect production of convective precipitation in the summer in the Arctic. Can you give also some possible reasons to explain the underestimation of BC in your paper (e.g. emissions, plume injection, numerical diffusion, aerosol scavenging, ...) and some ways of improvements ?

P16L5-8 : I cannot believe that missing plumes during one specific day (8 April 2015) could explain such a substantial discrepancy. Furthermore, writing "is perhaps due " is "perhaps" a bit too colloquial for a paper (but not a review !). I don't see what this sentence brings to the analysis.

P16L24 : What do you mean exactly by the "misrepresentation of these plumes" ? Give rather the physical processes responsible for the strong BC underestimation : emissions ? numerical diffusion ?

P17L12-13 : It does not seem right in summer. Why are the BC columns so low in July near northern Russia ? Are the flares off in summer ? Is it realistic ?

P17L17 : I would remove the word "vertical" and keep only the term "spatial" distributions.

P18L2-4 : This is not only caused by different transport patterns along which air masses reach the Arctic region, but also by different transport efficiencies due to scavenging.

P19L8 : I disagree with the use of the word "generally" here. A lot of studies have been focused on the same objective and they are not necessarily in agreement. How do the results of this study compare for example to Table A1 of Wang et al. (2014) who summarized twelve studies focusing on the source attribution of Arctic BC ?

P20L6-7 : Another possible explanation could be that biomass burning plumes have been transported at a too low altitude. This would explain the strong overestimation at the surface and the marked underestimation aloft. Is it a possible scenario ?

P23L1 : Here again, the simulated vertical BC profile is not consistent with that measured. The following sentence is a good way to say this.

P24L5-8 : This is an odd conclusive remark for a study focused on model simulations. It may be better to insist on the model uncertainties in the result mentioned above mentioned and how they can be addressed in future work.

Fig. 3 : What is the uncertainty on simulated monthly BC concentrations ? It could be estimated by the standard deviations of the model results used to derive the monthly averages. Please add them on Fig. 3 at least for the HTAP+flaring emission inventory.

Fig. 4 : Could you add an errorbar on the ground mean concentrations ?

Fig. 5 : This caption is good, but some of this text should also be reported in the manuscript, in particular L6-7 to understand how the analysis is performed.

Fig. 6 : This is not any flaring plume in July. Is is normal ?

Fig. 8 : Why aren't there error bars at Barrow and Ny Alesund stations as at Alert ?

Fig. 9 : Why don't you take into account 12 months to build the annual map ? I think this is confusing. It is fine to present maps for particular months (Jan, Apr, Sept) but I don't undertand the purpose of computing an annual average based on only 3 months, when 12 have actually been simulated.

Table 1 : Why don't you also include rRMSE in this stable ? The discussion relative to this section would be clearer.

—

Sharma, S., Ishizawa, M., Chan, D., Lavoué, D., Andrews, E., Eleftheriadis, K., and

Maksyutov, S.: 16-year simulation of Arctic black carbon: Transport, source contribution, and sensitivity analysis on deposition, Journal of Geophysical Research: Atmospheres, 118, 943–964, 2013.

Wang, H., Rasch, P. J., Easter, R. C., Singh, B., Zhang, R., Ma, P.-L., Qian, Y., Ghan, S. J., and Beagley, N.: Using an explicit emission tagging method in global modeling of source-receptor relationships for black carbon in the Arctic: Variations, sources, and transport pathways, Journal of Geophysical Research: Atmospheres, 119, 12,888–12,909, doi:10.1002/2014JD022297, http://dx.doi.org/10.1002/2014JD022297, 2014JD022297, 2014.

Raut, J.-C., Marelle, L., Fast, J., Thomas, J. L., Weinzierl, B., Law, K. S., Berg, L., Roiger, L., Easter, R., Heimerl, K., Onishi, T., Delanoe, J. and Schlager, H. : Cross-polar transport and scavenging of Siberian aerosols containing black carbon during the 2012 ACCESS summer campaign, Atmos. Chem. Phys. Disc., doi:10.5194/acp-2016-1023, 2017.

Breider, T. J., Mickley, L. J., Jacob, D. J., Wang, Q., Fisher, J. A., Chang, R., Alexander, B., et al.: Annual distributions and sources of Arctic aerosol components, aerosol optical depth, and aerosol absorption, Journal of Geophysical Research: Atmospheres, 119, 4107–4124, 2014.

Eckhardt, S., Quennehen, B., Oliviè, D. J. L., Berntsen, T. K., Cherian, R., Christensen, J. H., Collins, W., Crepinsek, S., Daskalakis, N., Flanner, M., et al.: Current model capabilities for simulating black carbon and sulfate concentrations in the Arctic atmosphere: a multi-model evaluation using a comprehensive measurement data set, Atmospheric Chemistry and Physics, 15, 9413–9433, 2015.

—

Technical comments :

In many places, add a space between value and unity, e.g. P2L6, P7L22, P8L4, P11L4,

P15L8, P15L16, P15L17, P16L1, P16L18, P16L26, P18L9, P18L22, P23L3, P23L11, P23L15.

P3L8 : Remove the dot before the parenthesis.

P7L21 : Neodymium dopped YAG is generally abbreviated as Nd:YAG, not Ni-YAG.

P9L25 : ACP journal recommends to write Figure instead of Fig. at the beginning of a sentence.

P10L6 : Replace semi-colon by comma.

P17 : section should be abbreviated as Sect. in ACP.

P13L10-15 : There is a typo in aethalometer L10, L11 and L15.

P22L4 : "to a half of the impact": this is not very clear. Please reformulate.

P27L10-14 : Evangeliou et al. (2016) : the paper has been published in ACP in June 2016. It is not a discussion paper anylonger.

---

## Referee Comment (RC2) · Anonymous Referee #2 · 13 May 2017

**Review of "Source attribution of Arctic black carbon constrained by aircraft and surface measurements," by Xu et al.**

In this paper, Xu and coauthors use the GEOS-Chem transport model to quantify the contributions from different regions to the Arctic black carbon burden during three years – 2009, 2011, and 2015. They first validate the model with surface-based monthly mean observations and with measurements from two springtime aircraft campaigns. They find relatively good agreement between the model and observed concentrations. For two Arctic sites (but not a third), this agreement improves when they include in their model an inventory of gas flaring emissions from western Siberia. Sensitivity studies with the forward model yield the contributions from different regions to Arctic BC, while simulations with the adjoint version of GEOS-Chem provide spatially-resolved information on these contributions.

The main findings of this paper are as follows: Anthropogenic BC from eastern and southern Asia dominate the Arctic BC burden in spring and contribute about one-third of the annual burden, with larger contributions aloft than near the surface. Anthropogenic BC from northern Asia are important BC in the lower troposphere, especially in spring. Biomass burning contributes 25% of Arctic BC annually. Results from the adjoint point to interesting influences on Arctic BC from regions as far south as the Indo-Gangetic Plain.

Main criticisms.

1. This paper moves forward the research on the origins of Arctic haze, providing in particular an update on how recent increases in anthropogenic BC from Asia may affect the Arctic. However, the authors do not make clear how their work builds on four recent GEOS-Chem studies that focus wholly or in part on Arctic BC: Wang et al. (2011, 2014) and Breider et al. (2014, 2017).

No doubt the authors were unaware of the 2017 paper, but the other three papers were published well before this one was submitted. Only Wang et al. (2011) is mentioned, and that only in passing. It is in particular concerning that the authors do not make clear whether they took advantage of the improvements in BC wet deposition of Wang et al. (2011, 2014). Did the authors include the snow scavenging scheme and the improvements to washout and rainout from Wang et al. (2011)? What about improvements to the impaction scavenging (Wang et al., 2014)? As is, the text cites only the wet deposition scheme of Liu et al. (2001). If the authors chose not to implement the Wang et al. (2011, 2014) improvements to wet deposition, the reader will want to know the rationale and what difference it would make if these improvements had, in fact, been included.

Brieder et al. (2014) focused on Arctic haze in 2008, and Brieder et al. (2017) examined the evolution of Arctic haze from 1980 to 2010. The authors can easily make the case that by simulating Arctic haze in 2009, 2011, and 2015, their paper provides an update to the Breider research, especially in light of increasing Asian emissions. But first they need to compare their approach and results very carefully with those in the earlier work. For example, the Brieder papers make use of a different emission inventory than does the current paper, and the reader will want to know how these inventories differ. As another example, Brieder et al. (2014)

appears to capture the mid-tropospheric peak in BC, while the current work does not. Again the reader will want to understand this discrepancy.

Responding to criticism #1 will require some effort. A close reading of the four relevant papers is necessary, and a detailed account of how the current paper moves beyond the previous papers is expected by the reader.

2. The conclusion section lacks discussion. Why should readers care about these new results? For example, what are the implications for their findings for regional climate in the Arctic? The introduction mentions some of the probable effects of BC on regional climate, and how the meteorological impacts of atmospheric BC likely differ with altitude. What does this altitude variation in forcing mean for Arctic haze of Asian origin? In addition, Brieder et al. (2017) suggests that the 1980-2010 trends in Arctic haze have contributed to regional warming. How do the new results build on Brieder et al. (2017)? How are emissions in Asia projected to change in the future, and what are the probable consequences for Arctic climate? Is gas flaring around the Arctic expected to ramp up in future decades?

3. The introduction lacks key information but is nonetheless too long. First, the authors should describe what is known about the seasonal variation of transport to the Arctic at the beginning of the paper. As is, this information appears scattered through the paper as a kind of recurring explanation for the modeled results. It would be easier for the reader to encounter this information in a succinct paragraph in the beginning, and then be reminded of how transport influences Arctic as the results emerge.

That said, the authors should condense much of the other background information in the introduction, beginning at line 14 on page 3 and continuing to the end of that section. For example, the reader doesn't need to know every published estimate of the influence of biomass burning on Arctic BC. Details of the Arctic aircraft campaigns can be saved for later in the paper.

4. The authors make much of recent increases in Asian BC emissions, but use anthropogenic emissions only for 2010 and GFED emissions for 2009, 2011, and 2014. These emissions are applied to GEOS-Chem simulations driven by 2009, 2011, and 2015 meteorological fields. The reader is curious if there are implications in using constant anthropogenic emissions and GFED emissions from a mismatched year. Also of interest is whether the authors see much interannual variation in transport over the three model years.

Minor criticisms.

Page 1, line 16. Run-on sentence.

Page 2, line 28. What is meant by "near-surface"?

Page 4, line 11. Reader is curious why published BC measurements may be biased.

Section 2.1. Years of measurements should be stated.

Page 9, line 3. The authors should consider a table providing BC emissions by region, as in Breider et al. (2014).

Page 12, line 25. Reader is confused why the measurements at Ny Alesund are halved.

Page 20, line 20. How "substantially" are shipping emissions expected to increase and over what time frame?

Page 20, line 25. The authors state: "The main difference is due to emission trends that our anthropogenic emissions from eastern and southern Asia are generally 30% higher than those in other studies." Are these increases due to increased development in Asia? Please remind the reader what time frame is being considered here.

Page 21, lines 12-21. Using the adjoint, the authors find that emissions as far south as the Indo-Gangetic Plain influence Arctic BC. This is new information. How confident are the authors of the GEOS-Chem simulation in this region (and in China)?

Figure 1. Are these total BC emissions or just anthropogenic?

Figure 3. Error bars on most measurements look very small. Please check the magnitudes. What are the years of the measurements?

Figure 4. Please put error bars on the ground-based measurements.

Figure 5. Please state in the caption the year and season of the measurements and model results.

Figure 7. Consider making a 4-panel plot with two new panels showing the stacked percent contribution of each region to the BC at different altitudes. The two new panels would have altitude on the y-axis, and percent contribution from 0-100% along the x-axis. In any case, the two existing panels look strangely elongated.

Figure 8. Measurements should have error bars.

Table 1. Table should include footnotes so that the reader does not have to scramble through the text to learn what the different scenarios mean. Also, it's not that clear that the vertical RMSE is meaningful since it varies so much with altitude.

**References.**

Breider, T.J., L.J. Mickley, D.J. Jacob, Q. Wang, J.A. Fisher, R.Y.-W. Chang, and B. Alexander (2014), Annual distributions and sources of Arctic aerosol components, aerosol optical depth and aerosol absorption, J. Geophys. Res. Atmos., 119, 4107-4124.

Breider, T. J., et al. (2017), Multidecadal trends in aerosol radiative forcing over the Arctic: Contribution of changes in anthropogenic aerosol to Arctic warming since 1980, J. Geophys. Res. Atmos., 122, 3573–3594.

Wang, Q., D. J. Jacob, J.A. Fisher, J. Mao, E.M. Leibensperger, C.C. Carouge, P. Le Sager, Y. Kondo, J.L. Jimenez, M.J. Cubison, and S.J. Doherty (2011), Sources of carbonaceous aerosols and deposited black carbon in the Arctic in winter-spring: implications for radiative forcing, Atmos. Chem. Phys., 11, 12,453-12,473.

Wang, Q., D. J. Jacob, J. R. Spackman, A. E. Perring, J. P. Schwarz, N. Moteki, E. A. Marais, C. Ge, J. Wang, and S. R. H. Barrett (2014), Global budget and radiative forcing of black carbon aerosol: Constraints from pole-to-pole (HIPPO) observations across the Pacific, J. Geophys. Res. Atmos., 119, 195–206.

---

## Referee Comment (RC3) · Anonymous Referee #3 · 24 May 2017

GENERAL COMMENTS The present paper describes results from air campaigns and an effort to understand the transport and origin of BC from different regions and emission sectors through modelling. The paper does not really include any new story information about the transport, missing sources or origin of BC to the Arctic. This is very obvious, because the authors frequently justify most of their sentences with references throughout the whole manuscript. So, what the authors claim in the present study has been already well described in previously published articles, although values are different. For instance, the contribution from Europe or Asia to BC in an Arctic station may differ in the present manuscript compared to other paper. However, this is still nothing new, as it may differ due to the use of different models or due to different lifetime of BC within each of the models or for any other reason that induces modelling uncertainty. Nevertheless, measurements in the Arctic are very useful and generally

lack and especially measurements from air campaigns. So, I would suggest that the authors should focus more on the measurements from the air campaigns and try to shorten the manuscript by removing the trivial statements about issues that have already been published elsewhere. Since the editor thinks that the present manuscript is novel enough to get out for a review, then I think that it deserves publication. It is very well written and it was a pleasure to reading it, although improvements can be applied, in order to be clearer and more concise. I could not find any weak point except those that I already pointed out. Everything flows well in it. Therefore, I only have some minor comments.

SPECIFIC COMMENTS Please shorten the Abstract. E.g., Page 2 – Line 9-10 is a trivial statement and can be removed from the abstract. Please follow this pattern and mention the most important points of your paper only and not all the conclusions!

P 5 – L 16: Should it be "state-of-the-art" instead of "state-of-the-science"?

P 5 – L 15 until the end of paragraph: You are describing methodology in the Introduction. Please remove all these details from this chapter!

P 5 – L 21 until the end of paragraph: Again you describe methodological issues that do not belong there, but rather in the next section of your paper.

P 6 – L 26: EMEP and WDCA are mentioned for the first time in the manuscript and need explanation. Please do the same elsewhere (e.g., SP2).

P16 – comments on Fig. 6: I had really hard times to follow this part and I think it is due to the poor labeling on the Figure. Therefore, I would suggest to put 6 small letters on each of the figures and point them in the text, so the reader knows to which of the figures you refer in the text.

P16 – L19-20 and L21: You are talking about the origin of the plume that arrives at the hotspot areas, but evidence is lacking. You have to point to respective figures somewhere or then remove these lines, because they cannot stand alone without any

justification.

P17 – L5-19: In my opinion column concentrations at the bottom panels of Fig.6 there do not say much. I think it is necessary to show the same maps with emissions. Preferably, add another panel (bottom) and show emissions in the same periods as with the column concentrations.

———————————————

---

## Author Comment (AC1) · 1 Jul 2017

We would like to thank Referee #1 for his/her useful comments and suggestions that helped to improve the quality of this manuscript. Reponses to these comments are provided below.

Specific Comments:

P1L15: I found the abstract too long, written like a conclusion section. Please try to make it more concise and underline the key messages from your study.

Response: Thanks for the suggestions. The abstract has been condensed.

P1L27-28: I think you oversell the performance of the model in reproducing the vertical

profile of BC. I don't think you can say that the simulations are "consistent" with the airborne measurements of BC. See for example Fig. 5 where the simulated profiles are flat, especially between 700 and 500 hPa, whereas the observations demonstrate a significant enhancement of BC concentrations. See more comments below (P15L14, P17L17 or P23L1).

Response: Thanks for pointing this out. We have clarified that our simulations underestimate BC concentrations in the middle troposphere in the abstract and the following paragraphs.

P6L4-7: The way it is phrased makes it difficult to understand what is new in this study. Do the authors develop the approach? Do they apply it to a different/longer period than that in Qi et al. (2017a)? Highlight the differences between the methodologies detailed in Kopacz et al. (2011) or Qi et al. (2017a).

Response: Qi et al. (2017a) has been revised to Qi et al. (2017b). We extended the time period from one month (April) in Qi et al. (2017b) to four months (each representing one season) and extended from the surface BC in Qi et al. (2017b) to the column BC that has implications for radiative forcing. We have highlighted these differences in the sentence as "We extend the application of this method to investigate the seasonal and annual responses of Arctic column BC to changes in regional emissions.". Kopacz et al. (2011) has been removed because this is a relatively old study and it not directly relevant to this study.

P6L8: Say that GEOS-Chem is a global CTM. It was only mentioned in the abstract.

Response: Done.

P7L5-6: Can you justify why no scattering corrections to the aethalometer measurements contrary to what is done for the PSAP data (P7L12)?

Response: Good question. The manufacturer's recommended MAC of 16.6 m2 g-1 is calibrated to account for multiple scattering (Sharma et al., 2017), thus no additional

scattering corrections are necessary. We have revised the description to the following in the manuscript: "This MAC value is recommended by the manufacturer for Model AE31 at 880 nm to account for absorption by BC and additional light scattering by both particles and filter fibers."

P10L11-16: How are fire emissions injected in the model? Are the emissions confined to the surface or does the model takes into account the variability in injection heights (function of burnt vegetation, intensity of fires, buoyancy, ...)?

Response: Fire emissions are injected into the boundary layer in the model. We have clarified this as "Biomass burning emissions are injected into the boundary layer in our simulations" in the manuscript.

P10L22: The description of the simulation is not given in this study. The authors give only a reference to an "old" paper (Park et al., 2003), which cannot be taken as a good reference to fully decribe what has been really done in this work. Please give additional details: period of the simulation, horizontal and vertical resolutions, . . . A lot of modifications have been included in the GEOS-Chem model from the study of Park et al. (2003). Some of them are likely to produce substantial changes in the BC distributions (e.g. wet deposition). Try to underline the main improvements.

Response: Thanks for the suggestion. We have included a detailed description of wet and dry deposition with major improvements since Park et al. (2003) in the model as the following in the manuscript:

"Dry deposition of BC aerosols adopts a standard resistance-in-series scheme as described in Zhang et al. (2001) with improvements on BC dry deposition velocity over snow and ice following Fisher et al. (2010) and Wang et al. (2011). Wet deposition of BC aerosols is initially described in Liu et al. (2001) and developed by Wang et al. (2011) to distinguish between liquid cloud (T > 268 K) in which 100 % hydrophilic BC is removed and ice cloud (T < 268 K) in which only hydrophobic BC is removed."

[Figure]

Additional details including the period of the simulation, horizontal and vertical resolutions are described on page 10 line 11-19.

P10L26: Does the model include wet deposition also in subgrid convective updrafts?

Response: Yes. This is described in Liu et al. (2001).

P13L8: The authors explain that EBC concentrations are biased high relative to rBC measurements because of absorbing components and errors in the coefficient used to derive BC concentrations from absorption coefficients. But the fact that EC concentrations are always larger than rBC is never explained. Is it similar to what was observed in other studies? Can it be caused by a difference in sampling the plumes?

Response: Thanks for point this out. Sharma et al. (2017) showed that some of the difference between EC and rBC could be explained by the presence of pyrolysis OC and carbonate carbon that might remain in aerosols after heating to 870 C in the thermal method but were removed in aerosols at 3600C in the refractory method. We have included this reason in the manuscript as "EC concentrations are lower than EBC concentrations from the Aethalometer, yet still high relative to rBC partly due to the presence of pyrolysis OC and carbonate carbon (Sharma et al., 2017)."

P13L14-16: May this discrepancy between aethalometer and PSAP measurements be also ascribed to differences in the calibration method: scattering corrections applied to PSAP but not to aethalometer data? And/or spectral dependency of the absorption coefficient of other substances than BC?

Response: The questions on scattering corrections were explained in the response to P7L5-6.

To understand the effect of spectral dependency of the absorption coefficient of other substances (i.e. brown carbon) on our results, we examined the Aethalometer EBC measurements at 370 nm at Barrow to see if the summer peak still exists at 370 nm. The result showed a distinct peak in July and August with a concentration increased by

10 ng m-3 compared to June and September, indicating the existence biomass burning. Thus, the summer peak in Aethalometer measurements at Barrow was not affected by the wavelength of measurements, but was influenced by biomass burning that was missing in PSAP measurements. We have revised the manuscript to the following to clarify this point.

"The summer peak is also observed in Aethalometer EBC measurements at 370 nm that is sensitive to brown carbon, indicating the influence of biomass burning. Unintentional exclusion of biomass burning plumes in the local pollution data screening performed for PSAP measurements at Barrow could contribute to the bias between the PSAP and the Aethalometer there (Stohl et al., 2006)."

P14L11-14: The peak observed in model simulations at Barrow is explained by a wrong timing of biomass burning emissions. Shouldn't it be the case for all models compared in the Eckhardt et al. (2015) paper if they all use the same fire emissions (with the same timing)? What is the role of injection heights in summer? And what is the contribution of wet deposition processes in the different models?

Response: Eckhardt et al. (2015) used identical biomass burning emissions (GFED 3.1) for all the models in their study. They attributed the difference of simulated surface BC concentrations from different models not only in summer but also the whole year to the treatment of aerosol wet scavenging in the models.

It is not clear what the effect of injection heights is on Arctic surface BC concentrations because some models (i.e. FLEXPART and HadGEM3) do not show the summer peak with biomass burning emissions distributed evenly within the boundary layer, while some other models (i.e. ECHAM6-HAM2) do show the summer peak with the same biomass burning emissions algorithm.

We have revised the sentence from "At Barrow all simulations show a distinct peak in July, which is due to the timing of biomass burning." to "At Barrow all simulations show a distinct peak in July, which is partly due to the timing of biomass burning."

P15L14: I think this sentence oversells the model prediction. The main feature noticed in the observed BC profile is the significant enhancement between 700 and 500 hPa, which is not captured at all by the simulations. So the model does not "generally represent "the relative vertical distribution of BC. The next sentence in the manuscript is a much better way to say this.

Response: We have revised the sentence to "All simulations generally represent the near constant vertical distribution of BC measurements from the surface to 700 hPa, and the decrease above 500 hPa, yet none represent the enhancement between 700-500 hPa."

P15L17-18: The way the sentence has been written gives the impression that most models also strongly underestimate BC in the mid or upper-troposphere. You cannot say this without also referring to more recent studies, highlighting either an overestimation aloft (e.g. Sharma et al. 2013 ; Wang et al., 2014), an improvement in the mid-troposphere (e.g. Breider al., 2014 ; Raut et al., 2017) or a very distinct behaviour between models (as in Eckhardt et al., 2015). In particular both Breider al. (2014) and Wang et al. (2014) used the GEOS-Chem model to represent BC in the Arctic and the results are not similar. This latter study suggested an incorrect production of convective precipitation in the summer in the Arctic. Can you give also some possible reasons to explain the underestimation of BC in your paper (e.g. emissions, plume injection, numerical diffusion, aerosol scavenging, ...) and some ways of improvements?

Response: Thanks for the suggestion. We have referred to Koch et al. (2009), Eckhardt et al. (2015), Breider et al. (2014) and Wang et al. (2011) in the discussion and have proposed some possible reasons for the underestimation as the following:

"The remaining underestimation of 14 ng m-3 RMSE in 500-700 hPa in the HTAP+flaring simulation is possibly due to insufficient magnitude or altitude comparisons of model with ARCTAS and ARCPAC measurements (Koch et al., 2009; Wang et al., 2011; Breider et al., 2014; Eckhardt et al., 2015) as proposed based on preferential

sampling by the aircraft of plumes discussed further below."

P16L5-8: I cannot believe that missing plumes during one specific day (8 April 2015) could explain such a substantial discrepancy. Furthermore, writing "is perhaps due" is "perhaps" a bit too colloquial for a paper (but not a review!). I don't see what this sentence brings to the analysis.

Response: We have rephrased this sentence to indicate the potential sampling bias of plumes and have given another possible reasons as described in the response to the previous comment.

P16L24: What do you mean exactly by the "misrepresentation of these plumes"? Give rather the physical processes responsible for the strong BC underestimation: emissions? numerical diffusion?

Response: By misrepresentation, we meant low in magnitude. Emissions or numerical diffusion might be responsible for the underestimation. We have revised the sentence to "The underestimated magnitudes of these plume, likely related to emissions or numerical diffusion, may contribute to the underestimation of BC concentrations between 500-700 hPa in Fig. 5".

P17L12-13: It does not seem right in summer. Why are the BC columns so low in July near northern Russia? Are the flares off in summer? Is it realistic?

Response: We have included emissions for Jan, Apr and Jul at the bottom panel of Fig. 6. As shown by the figure, flaring emissions in July are very similar to those in the other months. The minor flaring contribution in summer is shown not only for the column but also at the surface where the flaring contribution is usually the largest, as shown in the red shadings of Fig. 3, which is consistent with Stohl et al. (2013). Thus, the flares are not off in summer but the effective wet scavenging in summer likely removes most of BC from anthropogenic sources including flaring. We have included this explanation in the manuscript as the following:

"In July, the enhanced concentrations in western Siberia due to flaring are less obvious, due to more effective wet scavenging in summer."

P17L17: I would remove the word "vertical" and keep only the term "spatial" distributions.

Response: Done.

P18L2-4: This is not only caused by different transport patterns along which air masses reach the Arctic region, but also by different transport efficiencies due to scavenging.

Response: We have revised the sentence to "partly reflecting different transport pathways and scavenging efficiencies".

P19L8: I disagree with the use of the word "generally" here. A lot of studies have been focused on the same objective and they are not necessarily in agreement. How do the results of this study compare for example to Table A1 of Wang et al. (2014) who summarized twelve studies focusing on the source attribution of Arctic BC?

Response: Thanks for the suggestion. We have compared to Huang et al. (2010), Sharma et al. (2013), Shindell et al. (2008), Stohl (2006) and Ma et al. (2013) in Table A1 of Wang et al. (2014) and have revised the paragraph to the following

"The largest contribution from eastern and southern Asia to Arctic BC burden in this study is consistent with Ma et al. (2013) and Wang et al. (2014). However, some prior studies suggested that Europe had the largest contribution to Arctic BC burden (Stohl, 2006; Shindell et al., 2008; Huang et al., 2010a; Sharma et al., 2013) The difference likely arises from trends in anthropogenic emissions with reductions from Europe and increases in eastern and southern Asia as discussed further below."

P20L6-7: Another possible explanation could be that biomass burning plumes have been transported at a too low altitude. This would explain the strong overestimation at the surface and the marked underestimation aloft. Is it a possible scenario?

Response: In the response to P13L14-16, we have described that biomass burning plumes are likely missed by the PSAP measurements. Thus, it is not clear to us whether the discrepancy should be attributed to the simulation or the measurements.

P23L1: Here again, the simulated vertical BC profile is not consistent with that measured. The following sentence is a good way to say this.

Response: Revised.

P24L5-8: This is an odd conclusive remark for a study focused on model simulations. It may be better to insist on the model uncertainties in the result mentioned above mentioned and how they can be addressed in future work.

Response: Thanks for the suggestion. We have added "The considerable impact of emissions from China and Indo-Gangetic Plain on the Arctic deserves further investigation." to the conclusion.

Fig. 3: What is the uncertainty on simulated monthly BC concentrations? It could be estimated by the standard deviations of the model results used to derive the monthly averages. Please add them on Fig. 3 at least for the HTAP+flaring emission inventory.

Response: Done.

Fig. 4: Could you add an errorbar on the ground mean concentrations?

Response: Done. But only one year (2009) of ground measurement is available for Ny Alesund, so no error bar presents in the figure.

Fig. 5: This caption is good, but some of this text should also be reported in the manuscript, in particular L6-7 to understand how the analysis is performed.

Response: Done.

Fig. 6: This is not any flaring plume in July. Is it normal?

Response: There is flaring plume in July but the magnitude is smaller than the other
seasons because of more effective wet scavenging in summer. This has been described in more details in the response to P17L12-13.

Fig. 8: Why aren't there error bars at Barrow and Ny Alesund stations as at Alert?

Response: Thanks for pointing out this error. Corrected.

Fig. 9: Why don't you take into account 12 months to build the annual map? I think this is confusing. It is fine to present maps for particular months (Jan, Apr, Sept) but I don't undertand the purpose of computing an annual average based on only 3 months, when 12 have actually been simulated.

Response: We do not have a full year simulation with the adjoint model because it is time consuming. We assume the four months (Jan, Apr, Jul and Sept) are representative of four seasons. We have revised the caption to "The annual map is the average of contributions in January, April, July and September calculated with the adjoint model."

Table 1: Why don't you also include rRMSE in this stable? The discussion relative to this section would be clearer.

Response: We have included rRMSE in Table 2.

Technical comments: In many places, add a space between value and unity, e.g. P2L6, P7L22, P8L4, P11L4, P15L8, P15L16, P15L17, P16L1, P16L18, P16L26, P18L9, P18L22, P23L3, P23L11, P23L15.

Response: Done.

P3L8: Remove the dot before the parenthesis.

Response: Done.

P7L21: Neodymium dopped YAG is generally abbreviated as Nd:YAG, not Ni-YAG.

Response: Done.

P9L25: ACP journal recommends to write Figure instead of Fig. at the beginning of a

sentence.

Response: Done.

P10L6: Replace semi-colon by comma.

Response: Done.

P17: section should be abbreviated as Sect. in ACP.

Response: Done.

P13L10-15: There is a typo in aethalometer L10, L11 and L15.

Response: Done.

P22L4: "to a half of the impact": this is not very clear. Please reformulate.

Response: Done.

P27L10-14: Evangeliou et al. (2016): the paper has been published in ACP in June 2016. It is not a discussion paper any longer.

Response: Done.

References:

[revised manuscript text omitted]

---

## Author Comment (AC2) · 1 Jul 2017

We sincerely thank the Referee #2 for taking the time to review our paper and for providing constructive suggestions for improvement. Reponses to these comments are provided below.

In this paper, Xu and coauthors use the GEOS-Chem transport model to quantify the contributions from different regions to the Arctic black carbon burden during three years – 2009, 2011, and 2015. They first validate the model with surface-based monthly mean observations and with measurements from two springtime aircraft campaigns. They find relatively good agreement between the model and observed concentrations. For two Arctic sites (but not a third), this agreement improves when they include in their

[Figure]

model an inventory of gas flaring emissions from western Siberia. Sensitivity studies with the forward model yield the contributions from different regions to Arctic BC, while simulations with the adjoint version of GEOS-Chem provide spatially-resolved information on these contributions.

The main findings of this paper are as follows: Anthropogenic BC from eastern and southern Asia dominate the Arctic BC burden in spring and contribute about one-third of the annual burden, with larger contributions aloft than near the surface. Anthropogenic BC from northern Asia are important BC in the lower troposphere, especially in spring. Biomass burning contributes 25% of Arctic BC annually. Results from the adjoint point to interesting influences on Arctic BC from regions as far south as the Indo-Gangetic Plain.

Response: Thank you.

Main criticisms.

1. This paper moves forward the research on the origins of Arctic haze, providing in particular an update on how recent increases in anthropogenic BC from Asia may affect the Arctic. However, the authors do not make clear how their work builds on four recent GEOS-Chem studies that focus wholly or in part on Arctic BC: Wang et al. (2011, 2014) and Breider et al. (2014, 2017). No doubt the authors were unaware of the 2017 paper, but the other three papers were published well before this one was submitted. Only Wang et al. (2011) is mentioned, and that only in passing. It is in particular concerning that the authors do not make clear whether they took advantage of the improvements in BC wet deposition of Wang et al. (2011, 2014). Did the authors include the snow scavenging scheme and the improvements to washout and rainout from Wang et al. (2011)? What about improvements to the impaction scavenging (Wang et al., 2014)? As is, the text cites only the wet deposition scheme of Liu et al. (2001). If the authors chose not to implement the Wang et al. (2011, 2014) improvements to wet deposition, the reader will want to know the rationale and what difference it would make if these

improvements had, in fact, been included.

Brieder et al. (2014) focused on Arctic haze in 2008, and Brieder et al. (2017) examined the evolution of Arctic haze from 1980 to 2010. The authors can easily make the case that by simulating Arctic haze in 2009, 2011, and 2015, their paper provides an update to the Breider research, especially in light of increasing Asian emissions. But first they need to compare their approach and results very carefully with those in the earlier work. For example, the Brieder papers make use of a different emission inventory than does the current paper, and the reader will want to know how these inventories differ. As another example, Brieder et al. (2014) appears to capture the mid-tropospheric peak in BC, while the current work does not. Again the reader will want to understand this discrepancy.

Responding to criticism #1 will require some effort. A close reading of the four relevant papers is necessary, and a detailed account of how the current paper moves beyond the previous papers is expected by the reader.

Response: Thanks for these suggestions. We use version 10 of GEOS-Chem, which was the latest version available at the start of this work. Thus the wet deposition of Wang et al. (2011) was implemented in our simulation. We have clarified this in text. The developments of Wang et al. (2014) were not implemented into GEOS-Chem until version 11, and thus were not included here. Furthermore, these developments have little effect in the simulations of Arctic BC as indicated by sensitivity simulations in the supporting information of Wang et al. (2014).

This manuscript is not intended to be a follow-up study of Breider et al. (2014) or Breider et al. (2017). Instead, this is an independent project (hence different emission inventories and model parameters) with different objectives. Breider et al. (2014) and Breider et al. (2017) studied major near-term climate forcers including BC in the Arctic with an emphasis on their roles in Arctic warming, whereas we aim to interpret recent measurements to investigate geographical sources and their contributions to

Arctic BC. Thus providing updates to Breider et al.'s research is not our purpose. The different emission years of Breider et al. (2014) likely contribute to differences in the middle troposphere due to different biomass burning. However, we have included a table (Table 1) with detailed regional anthropogenic and biomass burning emissions, and have given possible reasons for the discrepancy in the middle troposphere as the following to help readers understand our simulations.

"The remaining underestimation of 14 ng m-3 RMSE in 500-700 hPa in the HTAP+flaring simulation is possibly due to insufficient magnitude or altitude comparisons of model with ARCTAS and ARCPAC measurements (Koch et al., 2009; Wang et al., 2011; Breider et al., 2014; Eckhardt et al., 2015) as proposed based on preferential sampling by the aircraft of plumes discussed further below."

2. The conclusion section lacks discussion. Why should readers care about these new results? For example, what are the implications for their findings for regional climate in the Arctic? The introduction mentions some of the probable effects of BC on regional climate, and how the meteorological impacts of atmospheric BC likely differ with altitude. What does this altitude variation in forcing mean for Arctic haze of Asian origin? In addition, Brieder et al. (2017) suggests that the 1980-2010 trends in Arctic haze have contributed to regional warming. How do the new results build on Brieder et al. (2017)? How are emissions in Asia projected to change in the future, and what are the probable consequences for Arctic climate? Is gas flaring around the Arctic expected to ramp up in future decades?

Response: Thanks for the suggestion. We have included more discussions in the conclusion as the following: "The increasing BC fraction from eastern and southern Asia at higher altitudes could have significant implications for the Arctic warming by extending the trend in increasing BC radiative forcing efficiency found by Breider et al. (2017) driven by strong increase with altitude of the direct radiative forcing of BC ( Zarzycki and Bond, 2010; Samset and Myhre, 2015). Besides, anthropogenic emissions of BC in southern Asia are projected to increase under several IPCC scenarios (Streets et

al., 2004; Bond et al., 2013). The climate implications of BC emissions within the Arctic are concerning given their disproportionate warming effects and the potential for increasing Arctic shipping activity as ice cover declines (Sand et al., 2013)."

3. The introduction lacks key information but is nonetheless too long. First, the authors should describe what is known about the seasonal variation of transport to the Arctic at the beginning of the paper. As is, this information appears scattered through the paper as a kind of recurring explanation for the modeled results. It would be easier for the reader to encounter this information in a succinct paragraph in the beginning, and then be reminded of how transport influences Arctic as the results emerge.

That said, the authors should condense much of the other background information in the introduction, beginning at line 14 on page 3 and continuing to the end of that section. For example, the reader doesn't need to know every published estimate of the influence of biomass burning on Arctic BC. Details of the Arctic aircraft campaigns can be saved for later in the paper.

Response: Thanks for the suggestion. We have included the description of transport to the Arctic in the introduction as the following and have condensed the other background information.

"Analysis of observations have revealed that Arctic BC is primarily transported from regions outside the Arctic (Klonecki et al., 2003; Stohl, 2006). In winter, northern Eurasia is the primary source where air masses are cold enough to penetrate the polar dome into the Arctic lower troposphere (Stohl, 2006). Air masses from the relatively warm mid-latitudes (i.e. North America and Asia) are forced to ascend above the polar dome to the Arctic middle and upper troposphere (Law and Stohl, 2007). In spring, the warming of the surface leads to higher potential temperature over the Arctic and the northward retreat of the polar dome, facilitating the transport of air masses from mid-latitude regions to the Arctic (Stohl, 2006). However, large uncertainties remain in sources and geographical contributions to Arctic BC that require additional interpretation of observations to address."

4. The authors make much of recent increases in Asian BC emissions, but use anthropogenic emissions only for 2010 and GFED emissions for 2009, 2011, and 2014. These emissions are applied to GEOS-Chem simulations driven by 2009, 2011, and 2015 meteorological fields. The reader is curious if there are implications in using constant anthropogenic emissions and GFED emissions from a mismatched year. Also of interest is whether the authors see much inter annual variation in transport over the three model years.

Response: The time frame for the "recent" increase in Asian BC emissions is from 2000s to 2010. We have clarified this as the following in the manuscript: "The main difference is due to emission trends such that our anthropogenic BC emissions from eastern and southern Asia are generally 30 % higher than those in earlier studies (e.g. Shindell et al., 2008; Sharma et al., 2013) due to rapid development since 2000 and that our anthropogenic BC emissions in Europe are half those in prior studies due to European emission controls.".

We assume no significant change of Asian BC emissions from 2010 to 2015 because Asian BC emission growth plateaued after 2010 (Crippa et al., 2016). We also assume that using GFED 2014 emissions for 2015 simulation has little influence on our results because no abnormal forest fires have been reported for 2014 and 2015. These assumptions have been included in the manuscript on page 8 line 12-13 and page 9 line 22-23.

We do not see much inter-annual variation in transport over the three model years because the simulated vertical profiles of 2009 and 2011 campaign years are similar to each other and that the contribution from eastern and southern Asia pattern remain similar. Both 2009 and 2011 profiles show uniform (coefficient of variance of 0.08 for 2009 and 0.13 for 2011) distribution below 700 hPa and larger variation above 700 hPa. The 2015 profile exhibits a distinct enhancement in the middle troposphere that

may be affected by plumes.

Minor criticisms.

Page 1, line 16. Run-on sentence.

Response: We have revised it to "Black carbon (BC) contributes to the Arctic warming, yet sources of Arctic BC and their geographic contributions remain uncertain".

Page 2, line 28. What is meant by "near-surface"?

Response: We have revised it to "Near-surface (< 1 km) BC particles".

Page 4, line 11. Reader is curious why published BC measurements may be biased. Section 2.1. Years of measurements should be stated.

Response: We have revised the sentence to "Furthermore, evidence is emerging that the BC observations to which many prior modeling studies compared may have been biased by 30 % (Sinha et al., accepted) or a factor of 2 (Sharma et al., 2017) due to other absorbing components in the atmospheric aerosol."

Years of measurements have been included in Sect. 2.1.

Page 9, line 3. The authors should consider a table providing BC emissions by region, as in Breider et al. (2014).

Response: Done.

Page 12, line 25. Reader is confused why the measurements at Ny Alesund are halved.

Response: The measurements at Ny-Ålesund were not halved by us. We have clarified this as "Restricting measurements to common years changes monthly means by less than 13 %, except for a 40 % change at Ny-Ålesund in April that arises from limited data coverage in common years since PSAP measurements for April at Ny-Ålesund is not available in 2009."

Page 20, line 20. How "substantially" are shipping emissions expected to increase and

over what time frame?

Response: We have clarified these in the manuscript as "This source is expected to increase by 16 % by 2050. (Winther et al., 2014)".

Page 20, line 25. The authors state: "The main difference is due to emission trends that our anthropogenic emissions from eastern and southern Asia are generally 30% higher than those in other studies." Are these increases due to increased development in Asia? Please remind the reader what time frame is being considered here.

Response: We have clarified these in the manuscript as "The main difference is due to emission trends that our anthropogenic emissions from eastern and southern Asia are generally 30 % higher than those in earlier studies (e.g. Shindell et al., 2008; Sharma et al., 2013) due to rapid development since 2000".

Page 21, lines 12-21. Using the adjoint, the authors find that emissions as far south as the Indo-Gangetic Plain influence Arctic BC. This is new information. How confident are the authors of the GEOS-Chem simulation in this region (and in China)?

Response: Emissions are a major source of uncertainty in the simulation of the contributions from the Indo-Gangetic Plain and China to the Arctic. The emissions in China and the Indo-Gangetic Plain in the HTAP v2 inventory originate from the MICS Asia inventory that represent the best estimate of emissions in Asia (Li et al., 2017). However, uncertainties still exist, so we suggested further investigations in the conclusion as the following: "The considerable impact of emissions from China and Indo-Gangetic Plain on the Arctic deserves further investigation."

Figure 1. Are these total BC emissions or just anthropogenic?

Response: We have revised the caption to "The colormap indicates annual total BC emissions averaged over 2009, 2011 and 2015 as used in the GEOS-Chem simulation."

Figure 3. Error bars on most measurements look very small. Please check the magnitudes. What are the years of the measurements?

Response: Error bars of measurements at Alert were not included for the clarity of the figure, but error bars of the best estimate of BC measurements (mean EC and rBC measurements) at Alert were included. The error bar magnitudes have been corrected at Barrow and Ny Alesund. Measurement years are included in the legend.

Figure 4. Please put error bars on the ground-based measurements.

Response: Done. But only one year (2009) ground measurement is available for Ny Alesund, so no error bar presents in the figure.

Figure 5. Please state in the caption the year and season of the measurements and model results.

Response: Done.

Figure 7. Consider making a 4-panel plot with two new panels showing the stacked percent contribution of each region to the BC at different altitudes. The two new panels would have altitude on the y-axis, and percent contribution from 0-100% along the x-axis. In any case, the two existing panels look strangely elongated.

Response: Thanks for the suggestion. We have changed the figure to the 4-panel plot as suggested.

Figure 8. Measurements should have error bars.

Response: Done.

Table 1. Table should include footnotes so that the reader does not have to scramble through the text to learn what the different scenarios mean. Also, it's not that clear that the vertical RMSE is meaningful since it varies so much with altitude.

Response: Thanks for the suggestion. We have included footnotes in Table 2. The vertical RMSE for simulations with different emissions shows the improvement with seasonal residential heating and flaring emissions in simulating vertical distributions of BC concentrations.

References:

Breider, T. J., Mickley, L. J., Jacob, D. J., Wang, Q., Fisher, J. A., Chang, R. Y.-W. and Alexander, B.: Annual distributions and sources of Arctic aerosol components, aerosol optical depth, and aerosol absorption, J. Geophys. Res. Atmos., 119(7), 4107–4124, doi:10.1002/2013JD020996, 2014.

Breider, T. J., Mickley, L. J., Jacob, D. J., Ge, C., Wang, J., Payer Sulprizio, M., Croft, B., Ridley, D. A., McConnell, J. R., Sharma, S., Husain, L., Dutkiewicz, V. A., Eleftheriadis, K., Skov, H. and Hopke, P. K.: Multidecadal trends in aerosol radiative forcing over the Arctic: Contribution of changes in anthropogenic aerosol to Arctic warming since 1980, J. Geophys. Res. Atmos., 122(6), 3573–3594, doi:10.1002/2016JD025321, 2017.

Crippa, M., Janssens-Maenhout, G., Dentener, F., Guizzardi, D., Sindelarova, K., Muntean, M., Van Dingenen, R. and Granier, C.: Forty years of improvements in European air quality: regional policy-industry interactions with global impacts, Atmos. Chem. Phys., 16(6), 3825–3841, doi:10.5194/acp-16-3825-2016, 2016.

Li, M., Zhang, Q., Kurokawa, J., Woo, J.-H., He, K., Lu, Z., Ohara, T., Song, Y., Streets, D. G., Carmichael, G. R., Cheng, Y., Hong, C., Huo, H., Jiang, X., Kang, S., Liu, F., Su, H. and Zheng, B.: MIX: a mosaic Asian anthropogenic emission inventory under the international collaboration framework of the MICS-Asia and HTAP, Atmos. Chem. Phys., 17(2), 935–963, doi:10.5194/acp-17-935-2017, 2017.

Wang, Q., Jacob, D. J., Fisher, J. A., Mao, J., Leibensperger, E. M., Carouge, C. C., Le Sager, P., Kondo, Y., Jimenez, J. L., Cubison, M. J. and Doherty, S. J.: Sources of carbonaceous aerosols and deposited black carbon in the Arctic in winter-spring: implications for radiative forcing, Atmos. Chem. Phys., 11(23), 12453–12473, doi:10.5194/acp-11-12453-2011, 2011.

Wang, H., Rasch, P. J., Easter, R. C., Singh, B., Zhang, R., Ma, P.-L., Qian, Y., Ghan, S. J. and Beagley, N.: Using an explicit emission tagging method in global

modeling of source-receptor relationships for black carbon in the Arctic: Variations, sources, and transport pathways, J. Geophys. Res. Atmos., 119(22), 12,888-12,909, doi:10.1002/2014JD022297, 2014

---

## Author Comment (AC3) · 1 Jul 2017

We thank anonymous referee #3 for the helpful suggestions and questions, which have led to valuable improvements in our manuscript. Reponses to these comments are provided below.

GENERAL COMMENTS

The present paper describes results from air campaigns and an effort to understand the transport and origin of BC from different regions and emission sectors through modelling. The paper does not really include any new story information about the transport, missing sources or origin of BC to the Arctic. This is very obvious, because the authors frequently justify most of their sentences with references throughout the

whole manuscript. So, what the authors claim in the present study has been already well described in previously published articles, although values are different. For instance, the contribution from Europe or Asia to BC in an Arctic station may differ in the present manuscript compared to other paper. However, this is still nothing new, as it may differ due to the use of different models or due to different lifetime of BC within each of the models or for any other reason that induces modelling uncertainty. Nevertheless, measurements in the Arctic are very useful and generally lack and especially measurements from air campaigns. So, I would suggest that the authors should focus more on the measurements from the air campaigns and try to shorten the manuscript by removing the trivial statements about issues that have already been published elsewhere. Since the editor thinks that the present manuscript is novel enough to get out for a review, then I think that it deserves publication. It is very well written and it was a pleasure to reading it, although improvements can be applied, in order to be clearer and more concise. I could not find any weak point except those that I already pointed out. Everything flows well in it. Therefore, I only have some minor comments.

Response: Thanks for these. Novelties of this paper include 1) interpretation of new airborne measurements at Alert in the Arctic, 2) the first comparison with a chemical transport model of rBC measrements at Alert, 3) more accurate surface measurements used for model evaluation and source attribution, 4) improved understanding of how differet emission inventories affect comparison with observations, 5) source attribution using the adjoint of the GEOS-Chem model to understand the importance of specific sources, and 6) identification of the Tarim oilfield and Indo-Gangetic plain as important sources. We have revised the manuscript to highlight these novelties and have condensed less novel material.

SPECIFIC COMMENTS

Please shorten the Abstract. E.g., Page 2 – Line 9-10 is a trivial statement and can be removed from the abstract. Please follow this pattern and mention the most important points of your paper only and not all the conclusions!

[Figure]

Response: Done.

P 5 – L 16: Should it be "state-of-the-art" instead of "state-of-the-science"?

Response: Revised.

P 5 – L 15 until the end of paragraph: You are describing methodology in the Introduction. Please remove all these details from this chapter!

Response: Done.

P 5 – L 21 until the end of paragraph: Again you describe methodological issues that do not belong there, but rather in the next section of your paper.

Response: We have condensed this paragraph but we think it deserves a brief description of the motivation to use this method in the introduction because the adjoint of the GEOS-Chem simulation results are a highlight of this manuscript.

P 6 – L 26: EMEP and WDCA are mentioned for the first time in the manuscript and need explanation. Please do the same elsewhere (e.g., SP2).

Response: We have written out EMEP, WDCA and SP2 where they appear for the first time in the manuscript

P16 – comments on Fig. 6: I had really hard times to follow this part and I think it is due to the poor labeling on the Figure. Therefore, I would suggest to put 6 small letters on each of the figures and point them in the text, so the reader knows to which of the figures you refer in the text.

Response: Done.

P16 – L19-20 and L21: You are talking about the origin of the plume that arrives at the hotspot areas, but evidence is lacking. You have to point to respective figures somewhere or then remove these lines, because they cannot stand alone without any justification.

[Figure]

Response: Origin removed.

P17 – L5-19: In my opinion column concentrations at the bottom panels of Fig.6 there do not say much. I think it is necessary to show the same maps with emissions. Preferably, add another panel (bottom) and show emissions in the same periods as with the column concentrations.

Response: Thanks for the suggestion. We have included emissions at the bottom panel of Fig. 6.

---

## Referee Report (RR1)

**Second review of "Source attribution of Arctic black carbon constrained by aircraft and surface measurements," by Xu et al.**

In this paper, Xu and coauthors use the GEOS-Chem transport model to quantify the contributions from different regions to the Arctic black carbon burden during three years – 2009, 2011, and 2015. They first validate the model with surface-based monthly mean observations and with measurements from two springtime aircraft campaigns. They find relatively good agreement between the model and observed concentrations. For two Arctic sites (but not a third), this agreement improves when they include in their model an inventory of gas flaring emissions from western Siberia. Sensitivity studies with the forward model yield the contributions from different regions to Arctic BC, while simulations with the adjoint version of GEOS-Chem provide spatially-resolved information on these contributions.

In the first revision of their paper, the authors have addressed most of the comments. The plots have improved, and the introduction and conclusion both function much better.

**Main criticism.**
The authors did not respond adequately to Main Criticism #1 in my first review. That comment asked the authors to say more about how their study built on the Wang et al. (2011) and Breider et al. (2014, 2017) studies, which also investigated black carbon and its trends in the Arctic using GEOS-Chem. The authors responded:

*This manuscript is not intended to be a follow-up study of Breider et al. (2014) or Breider et al. (2017). Instead, this is an independent project (hence different emission inventories and model parameters) with different objectives. Breider et al. (2014) and Breider et al. (2017) studied major near-term climate forcers including BC in the Arctic with an emphasis on their roles in Arctic warming, whereas we aim to interpret recent measurements to investigate geographical sources and their contributions to Arctic BC.*

Three papers, all with the intent to validate GEOS-Chem BC in the Arctic, are not in fact "independent projects." Readers will try very hard to synthesize the results from these papers, and the authors of this paper should make that synthesis easier.

Indeed, a key goal of both Breider 2014 and Breider 2017 was to validate the GEOS-Chem simulation of Arctic BC. Validation was considered essential in the Breider papers; otherwise the radiative forcing calculated would not have been credible. Thus, readers will want to know how the new BC results differ from those of Breider and why. They will expect the current paper to compare emission inventories and model parameters with those used by Breider. Otherwise, what will the next GEOS-Chem user – or any chemistry modeler – do when she wants to model the Arctic atmosphere? What lessons can be learned? This comparison is especially crucial given the large difficulties current chemistry models have in simulating Arctic PM.

For example, Figures 3 and 4 of Breider 2014 reveals that adding gas flaring could indeed improve the model match with surface observations in that paper. But Breider 2014 better captures peak BC concentrations at ~5 km in spring than do any simulations in the new paper. Why is that? Is it just because of fires (Wang et al., 2011)? Or are there differences in wet

deposition schemes that matter? A key conclusion of the Xu paper is that "anthropogenic emissions in eastern and southern Asia have the largest effect on the Arctic BC column burden in spring (56%)...., with the largest contribution in the middle troposphere (400-700 hPa)." If that is the case, it matters that Breider 2014 captures the BC enhancement in the mid-troposphere but the new paper does not.

The authors also state:
*The developments of Wang et al. (2014) were not implemented into GEOS-Chem until version 11, and thus were not included here. Furthermore, these developments have little effect in the simulations of Arctic BC as indicated by sensitivity simulations in the supporting information of Wang et al. (2014).*

The authors should not assume that everyone knows that the developments in Wang 2014 were not implemented until v11 and in any would have little effect on Arctic BC. A key piece of writing any paper is to acknowledge what the current study lacks and then say whether or not that lack matters.

To describe the underestimate of the BC simulation in the mid-troposphere, the authors have added the following text:

*The remaining underestimation of 14 ng m-3 RMSE in 500-700 hPa in the HTAP+flaring simulation is possibly due to insufficient magnitude or altitude comparisons of model with ARCTAS and ARCPAC measurements (Koch et al., 2009; Wang et al., 2011; Breider et al., 2014; Eckhardt et al., 2015) as proposed based on preferential sampling by the aircraft of plumes discussed further below.*

The reader is confused by "insufficient magnitude." What exactly has insufficient magntitude? The wording of the entire sentence is awkward.

---

## Author Response (AR2)

**We sincerely thank the Referee #2 for reviewing our paper for the second time and for providing constructive suggestions for further improvement. Reponses to these comments are provided below.**

In this paper, Xu and coauthors use the GEOS-Chem transport model to quantify the contributions from different regions to the Arctic black carbon burden during three years – 2009, 2011, and 2015. They first validate the model with surface-based monthly mean observations and with measurements from two springtime aircraft campaigns. They find relatively good agreement between the model and observed concentrations. For two Arctic sites (but not a third), this agreement improves when they include in their model an inventory of gas flaring emissions from western Siberia. Sensitivity studies with the forward model yield the contributions from different regions to Arctic BC, while simulations with the adjoint version of GEOS-Chem provide spatially-resolved information on these contributions.

In the first revision of their paper, the authors have addressed most of the comments. The plots have improved, and the introduction and conclusion both function much better.

**Response: Thank you.**

Main criticism.

The authors did not respond adequately to Main Criticism #1 in my first review. That comment asked the authors to say more about how their study built on the Wang et al. (2011) and Breider et al. (2014, 2017) studies, which also investigated black carbon and its trends in the Arctic using GEOS-Chem. The authors responded:

*This manuscript is not intended to be a follow-up study of Breider et al. (2014) or Breider et al. (2017). Instead, this is an independent project (hence different emission inventories and model parameters) with different objectives. Breider et al. (2014) and Breider et al. (2017) studied major near-term climate forcers including BC in the Arctic with an emphasis on their roles in Arctic warming, whereas we aim to interpret recent measurements to investigate geographical sources and their contributions to Arctic BC.*

Three papers, all with the intent to validate GEOS-Chem BC in the Arctic, are not in fact "independent projects." Readers will try very hard to synthesize the results from these papers, and the authors of this paper should make that synthesis easier.

Indeed, a key goal of both Breider 2014 and Breider 2017 was to validate the GEOS-Chem simulation of Arctic BC. Validation was considered essential in the Breider papers; otherwise the radiative forcing calculated would not have been credible. Thus, readers will want to know how the new BC results differ from those of Breider and why. They will expect the current paper to compare emission inventories and model parameters with those used by Breider. Otherwise, what will the next GEOS-Chem user – or any chemistry modeler – do when she wants to model the Arctic atmosphere? What lessons can be learned? This comparison is especially crucial given the large difficulties current chemistry models have in simulating Arctic PM.

Response: Thank you for clarifying your perspective. We respectfully admit concern that undue attention is being placed on simulations from a single group, rather than on observations, on other GEOS-Chem simulations of the Arctic, or Arctic simulations by other models. Nonetheless, to address the reviewer's comments we have added additional citations of Wang et al. (2011) and Breider et al. (2014, 2017). We have carefully reread Breider et al. (2017) to follow their example in the discussion of differences from Breider et al. (2014). We now have 15 citations to these three references. Below is a list of text where Wang et al's and Breider et al's works are cited to provide evidence of how this study builds on those works.

Page 3 line 23: Some studies suggested that Europe was the dominant source of BC aloft (Stohl, 2006; Huang et al, 2010b) while others found eastern and southern Asia was the most important source (Sharma et al., 2013; Breider et al., 2014; Wang et al., 2014a; Ikeda et al., 2017) in the middle troposphere.

Page 4 line 6: BC emissions in mid- and low-latitude regions increase the Arctic climate forcing efficiency by altering the BC vertical distribution (Breider et al., 2017).

Page 5 line 3: Our work builds on knowledge gained from previous GEOS-Chem studies of Arctic BC (Wang et al., 2011; Breider et al., 2014; Breider et al., 2017; Qi et al., 2017a; Qi et al., 2017b) with major improvements including 1) new airborne measurements during 2009, 2011 and 2015 when more typical fires than in previous studies foster better understanding of anthropogenic source contributions to the Arctic; 2) new refractory BC measurements in the Arctic more accurately constrain emissions in simulations; 3) more recent and improved emissions better represent the global redistribution of BC emissions, include flaring and seasonal emissions of residential heating; and 4) seasonal source attribution using the adjoint of GEOS-Chem reveals the importance of specific sources.

Page 7 line 12: Prior Arctic aircraft campaigns (i.e. ARCTAS) were strongly influenced by the unusually extensive Russian fires in 2008 (e.g. Warneke et al., 2009; Wang et al., 2011; Breider et al., 2014). This study uses new aircraft observations when fires were less pronounced over multiple years (2009, 2011 and 2015) to better understand anthropogenic source contributions.

Page 8 line 24: The Bond et al. (2007) emission inventory for 2000 is included for comparison, since it has been widely used in modeling studies of Arctic BC (Shindell et al., 2008; Koch et al., 2009; Liu et al., 2011; Wang et al., 2011; Breider et al., 2014; Qi et al., 2017a; Qi et al., 2017b).

Page 10 line 1: Dry deposition of BC aerosols adopts a standard resistance-in-series scheme as described in Zhang et al. (2001) with improvements on BC dry deposition velocity over snow and ice following Fisher et al. (2010) and Wang et al. (2011). Wet deposition of BC aerosols is initially described in Liu et al. (2001) and developed by Wang et al. (2011) to distinguish between liquid cloud (T > 268 K) in which 100 % hydrophilic BC is removed and ice cloud (T < 268 K) in which only hydrophobic BC is removed.

Page 10 line 6: The scavenging developments of Wang et al. (2014b) are not implemented since they have little effect on Arctic BC.

Page 10 line 15: The time period simulated is 2009, 2011 and 2015, which is coincident with aircraft measurements when fires were more typical than for previous evaluations of GEOS-Chem versus

Arctic observations (i.e., Wang et al., 2011; Breider et al., 2014) to better understand anthropogenic source contributions here.

Page 14 line 11: This vertical distribution is similar to the measurements of the ARCTAS aircraft campaign in the Arctic in spring 2008 (Wang et al., 2011), though the magnitude of concentrations in this work is lower by a factor of about 2, likely because the Arctic was substantially influenced by strong biomass burning in northern Eurasia during the ARCTAS in spring 2008 (Warneke et al., 2009).

Page 21 line 13: The increasing BC fraction from eastern and southern Asia at higher altitudes could have significant implications for Arctic warming by extending the trend in increasing BC radiative forcing efficiency found by Breider et al. (2017) driven by strong increase with altitude of the direct radiative forcing of BC (Zarzycki and Bond, 2010; Samset and Myhre, 2015).

For example, Figures 3 and 4 of Breider 2014 reveals that adding gas flaring could indeed improve the model match with surface observations in that paper. But Breider 2014 better captures peak BC concentrations at ~5 km in spring than do any simulations in the new paper. Why is that? Is it just because of fires (Wang et al., 2011)? Or are there differences in wet deposition schemes that matter? A key conclusion of the Xu paper is that "anthropogenic emissions in eastern and southern Asia have the largest effect on the Arctic BC column burden in spring (56%)...., with the largest contribution in the middle troposphere (400-700 hPa)." If that is the case, it matters that Breider 2014 captures the BC enhancement in the mid-troposphere but the new paper does not.

Response: The mid-tropospheric peak in Breider et al. (2014) and in our work are not directly comparable due to different reasons for the peak (unusual biomass burning in Breider, possible sampling bias in this study). Furthermore, the underestimate in this study is minor (several percent). These topics are discussed further below.

One major difference in the simulation between Breider et al. (2014) and this study is fire emissions. A comparison to fire emissions in Breider et al. (2014) is confounded by the extensive tuning of that simulation to the unusual conditions of spring 2008 when boreal fires in Eurasia were unusually extensive. For example, Breider et al. (2014) scaled FLAMBE emissions from Russia by 47%, from southeast Asia by 55%, and from North America by 37.5%. The tuning used by Breider et al. (2014) is not applicable to our work since that tuning targeted the unusually extensive fire emissions in spring 2008.

Another major difference is anthropogenic emissions. Breider et al., (2014) used the Bond et al. (2007) emission inventory for 2000, with doubled emissions in Russia and Asia, while we use the HTAP emission inventory for 2010. The advantages of using the HTAP inventory over the Bond et al. (2007) inventory include 1) seasonally varying emissions of residential heating and 2) higher emissions in eastern and southern Asia that reflects the considerable growth of energy consumption in Asia over the past decades. These advantages are described in the manuscript on page 8 line 11 – 24. By using the HTAP inventory, we find that the mid-tropospheric burden is primary contributed by anthropogenic emissions in eastern and southern Asia during years (2009, 2011 and 2015) without abnormal fire activities.

Given the different reasons for the peak in the mid-troposphere (unusually extensive fires in Russia in Brieder et al. (2014); anthropogenic emissions from eastern and southern Asia in this study), we

**believe these two studies are not comparable in this aspect.**

**Furthermore, we respectfully contend that Breider et al. (2014) did not better capture the BC peak in the middle troposphere. Figure 4 of Breider et al. (2014) showed an overestimation of BC concentrations at 4 - 6 km by 30 - 50 ng m$^{-3}$, while we underestimate the BC concentrations at 700 – 500 hPa by 10 – 30 ng m$^{-3}$.**

**We reiterate that the weak 'peak' in this study could be influenced by preferential sampling of plumes by the aircraft as stated on page 15.**

**Finally, the 'peak' raised by the reviewer is a minor issue because the difference of Arctic BC burden below 500 hPa from the simulated and the observed vertical profile is as low as 6.5%. We have added this point on page 15 line 3 as the following:**
**"If the measurements are representative in this region, the Arctic BC burden below 500 hPa in springtime could be 6.5 % larger than simulated here."**

The authors also state:
*The developments of Wang et al. (2014) were not implemented into GEOS-Chem until version 11, and thus were not included here. Furthermore, these developments have little effect in the simulations of Arctic BC as indicated by sensitivity simulations in the supporting information of Wang et al. (2014).*

The authors should not assume that everyone knows that the developments in Wang 2014 were not implemented until v11 and in any would have little effect on Arctic BC. A key piece of writing any paper is to acknowledge what the current study lacks and then say whether or not that lack matters.

**Response: Thank you. We have clarified this in the manuscript on page 10 by adding the following "The scavenging developments of Wang et al. (2014b) are not implemented since they have little effect on Arctic BC."**

To describe the underestimate of the BC simulation in the mid-troposphere, the authors have added the following text:
*The remaining underestimation of 14 ng m-3 RMSE in 500-700 hPa in the HTAP+flaring simulation is possibly due to insufficient magnitude or altitude comparisons of model with ARCTAS and ARCPAC measurements (Koch et al., 2009; Wang et al., 2011; Breider et al., 2014; Eckhardt et al., 2015) as proposed based on preferential sampling by the aircraft of plumes discussed further below.*

The reader is confused by "insufficient magnitude." What exactly has insufficient magntitude? The wording of the entire sentence is awkward.

**Response: Thank you. We have revised the sentence to "The remaining underestimation of 14 ng m$^{-3}$ RMSE in 500-700 hPa in the HTAP+flaring simulation is possibly due to insufficient emissions or preferential sampling of plumes by the aircraft as discussed further below."**

**References:**

Bond, T. C., Bhardwaj, E., Dong, R., Jogani, R., Jung, S., Roden, C., Streets, D. G. and Trautmann, N. M.: Historical emissions of black and organic carbon aerosol from energy-related combustion, 1850-2000, Global Biogeochem. Cycles, 21(2), GB2018, doi:10.1029/2006GB002840, 2007.

Breider, T. J., Mickley, L. J., Jacob, D. J., Wang, Q., Fisher, J. A., Chang, R. Y.-W. and Alexander, B.: Annual distributions and sources of Arctic aerosol components, aerosol optical depth, and aerosol absorption, J. Geophys. Res. Atmos., 119(7), 4107–4124, doi:10.1002/2013JD020996, 2014.

Breider, T. J., Mickley, L. J., Jacob, D. J., Ge, C., Wang, J., Payer Sulprizio, M., Croft, B., Ridley, D. A., McConnell, J. R., Sharma, S., Husain, L., Dutkiewicz, V. A., Eleftheriadis, K., Skov, H. and Hopke, P. K.: Multidecadal trends in aerosol radiative forcing over the Arctic: Contribution of changes in anthropogenic aerosol to Arctic warming since 1980, J. Geophys. Res. Atmos., 122(6), 3573–3594, doi:10.1002/2016JD025321, 2017.

Koch, D., Schulz, M., Kinne, S., Mcnaughton, C., Spackman, J. R., Balkanski, Y., Bauer, S. and Berntsen, T.: Physics Evaluation of black carbon estimations in global aerosol models, Atmos. Chem. Phys., 9001–9026, 2009.

Liu, J., Fan, S., Horowitz, L. W. and Levy, H.: Evaluation of factors controlling long-range transport of black carbon to the Arctic, J. Geophys. Res., 116(D4), D04307, doi:10.1029/2010JD015145, 2011.

Qi, L., Li, Q., Li, Y. and He, C.: Factors controlling black carbon distribution in the Arctic, Atmos. Chem. Phys., 17(2), 1037–1059, doi:10.5194/acp-17-1037-2017, 2017a.

Qi, L., Li, Q., Henze, D. K., Tseng, H.-L. and He, C.: Sources of springtime surface black carbon in the Arctic: an adjoint analysis for April 2008, Atmos. Chem. Phys., 17, 9697-9716, doi.org/10.5194/acp-17-9697-2017, 2017b.

Shindell, D. T., Chin, M., Dentener, F., Doherty, R. M., Faluvegi, G., Fiore, a M., Hess, P., Koch, D. M., MacKenzie, I. a., Sanderson, M. G., Schultz, M. G., Schulz, M., Stevenson, D. S., Teich, H., Textor, C., O.Wild, Bergmann, D. J., Bey, I., Bian, H., Cuvelier, C., Duncan, B. N., Folberth, G., Horowitz, L. W., Jonson, J., Kaminski, J. W., Marmer, E., Park, R., Pringle, K. J., Schroeder, S., Szopa, S., Takemura, T., Zeng, G., Keating, T. J. and Zuber, a.: A multi-model assessment of pollution transport to the Arctic, Atmopsheric Chem. Phys., 8, 5353–5372, doi:10.5194/acp-8-5353-2008, 2008.

Wang, Q., Jacob, D. J., Fisher, J. A., Mao, J., Leibensperger, E. M., Carouge, C. C., Le Sager, P., Kondo, Y., Jimenez, J. L., Cubison, M. J. and Doherty, S. J.: Sources of carbonaceous aerosols and deposited black carbon in the Arctic in winter-spring: implications for radiative forcing, Atmos. Chem. Phys., 11(23), 12453–12473, doi:10.5194/acp-11-12453-2011, 2011.

Wang, Q., Jacob, D. J., Spackman, J. R., Perring, A. E., Schwarz, J. P., Moteki, N., Marais, E. A., Ge, C., Wang, J. and Barrett, S. R. H.: Global budget and radiative forcing of black carbon aerosol: Constraints from pole-to-pole (HIPPO) observations across the Pacific, J. Geophys. Res. Atmos., 119(1), 195–206, doi:10.1002/2013JD020824, 2014b.